# Transcriptomic investigation of the interaction between a biocontrol yeast, *Papiliotrema terrestris* strain PT22AV, and the postharvest fungal pathogen *Penicillium expansum* on apple

Giuseppe Ianiri ✉, Giuseppe Barone, Davide Palmieri, Michela Quiquero, Ilenia Gaeta, Filippo De Curtis & Raffaello Castoria ✉

Biocontrol strategies offer a promising alternative to control plant pathogens achieving food safety and security. In this study we apply a RNAseq analysis during interaction between the biocontrol agent (BCA) *Papiliotrema terrestris*, the pathogen *Penicillium expansum*, and the host *Malus domestica*. Analysis of the BCA finds overall 802 upregulated DEGs (differentially expressed genes) when grown in apple tissue, with the majority being involved in nutrients uptake and oxidative stress response. This suggests that these processes are crucial for the BCA to colonize the fruit wounds and outcompete the pathogen. As to *P. expansum* analysis, 1017 DEGs are upregulated when grown in apple tissue, with the most represented GO categories being transcription, oxidation reduction process, and transmembrane transport. Analysis of the host *M. domestica* finds a higher number of DEGs in response to the pathogen compared to the BCA, with overexpression of genes involved in host defense signaling pathways in the presence of both of them, and a prevalence of pattern-triggered immunity (PTI) and effector-triggered immunity (ETI) only during interaction with *P. expansum*. This analysis contributes to advance the knowledge on the molecular mechanisms that underlie biocontrol activity and the tritrophic interaction of the BCA with the pathogen and the host.

Nowadays food losses represent a critical issue that is strongly exacerbated by climate change[1]. In order to address such a global challenge, ongoing research is focused on the comprehension of molecular, epidemiological and ecological bases of plant diseases with the aim of developing effective and long-lasting alternative solutions that could replace or integrate the control strategies based on synthetic fungicides. One of the most promising approaches is biological control, which is defined as the mitigation of pests and diseases through the use of naturally occurring antagonists named biocontrol agents (BCAs), including bacteria, yeasts and filamentous fungi. Although there are many examples of potential BCAs that are able to decrease pathogens infections, there is still limited knowledge on their mode of action and, as a consequence, practical applications of BCAs in organic agriculture are still limited. Understanding the mechanisms conferring biocontrol activity is indeed the foundation to promote the use of the BCAs, from the development to their applications[2].

Yeast BCAs are widely studied against postharvest pathogens, and at present a number of potential yeast BCAs have been reported in the literature. The most common mechanisms of action reported are: competition for nutrients and space, secretion of enzymes cleaving polymers of pathogens cell wall, toxins production, release of volatile organic compounds (VOCs), mycoparasitism, and induction of resistance. In general, more than one mechanism of action may take place at the same time, although competition for nutrients and space is the main mechanism of yeast BCAs and it exploits the natural nutritional requirements of these beneficial microbes[2].

Department of Agricultural, Environmental and Food Sciences, University of Molise, via F. De Sanctis snc, 86100 Campobasso, Italy.
✉e-mail: giuseppe.ianiri@unimol.it; castoria@unimol.it

The BCA object of the present study is *Papiliotrema terrestris*, a basidiomycete yeast belonging to the order Tremellales within the subphylum Agaricomycotina. The strain that we used in this study is PT22AV. *Papiliotrema terrestris* strain PT22AV (TradeName YSY®) is a yeast biocontrol agent developed by AgroVentures srl (Italy) and AgroVentures LLC (USA) (www.agroventures.eu) to control a broad range of plant pathogens in multiple crops. YSY agronomical applications are patented and patent pending globally, and it is currently undergoing regulatory approval process in the European Union and in the USA. YSY applications are focused at replacing or integrating the use of chemical fungicides on multiple crops and pathogens, at critical plant growth stages both in the soil and on plant aerial parts. Product seems particularly fit to reduce impact on pollinators and before crop harvest to reduce chemical residues on fruits and vegetables.

The biocontrol activity of *P. terrestris* against *P. expansum* has been well characterized both at room temperature and during cold storage on different apple cultivars[3–6]. In particular, these studies revealed that *P. terrestris* displays elevated antagonistic activity also against other pathogens (*Botrytis cinerea*, *Rhizopus stolonifer* and *Aspergillus niger*) on different fruits (apples, pears, strawberries, kiwi fruits and table grapes) during storage at room temperature[5] and also in open field in the orchard against *Monilinia* spp. on stone fruits[7]. Mechanistic studies revealed that the main mode of action of *P. terrestris* is based on competition for nutrients and space, which relies on the rapid colonization of fruit wounds (wound competence), the main penetration sites of pathogens within host tissues. Elevated resistance to reactive oxygen species[(ROS: superoxide anion($O_2^-$) and hydrogen peroxide ($H_2O_2$)] generated by plant tissues as a consequence of wounding is critical for the wound competence of *P. terrestris* as demonstrated through enzymatic and genetic approaches. In particular, Castoria et al.[6] showed that the application of the ROS-deactivating enzymes superoxide dismutase and catalase in combination with the BCA resulted in a higher antagonistic activity against *B. cinerea* and *P. expansum* in apple fruit (cv Annurca) compared to a treatment in which the BCA was applied alone[6]. In agreement, a *P. terrestris* deletion mutant for the gene *YAP1*, a critical transcription factor required for general resistance to oxidative stress, displayed lowed antagonistic activity than the parental strain against *M. fructigena* and *P. expansum* on stored apple[8]. Moreover, other mechanisms could also play a role in the biocontrol activity of *P. terrestris*, such as the in vitro production of extracellular β-1,3-glucanase, an enzyme that depolymerizes fungal cell walls, when the BCA was grown in the presence of hyphal cell walls of *B. cinerea* and *P. expansum*, even though direct interaction with the pathogen hyphae was not detected[4]; and the ability to stimulate the growth of roots and shoots in maize grains and to increase the concentration of macro- and micro-nutrients in plant roots and shoots likely through the production of the plant hormone IAA (indole-3-acetic acid)[9].

The recent advent of the next generation sequencing (NGS) for applications of genomics and transcriptomics, combined with tools for gene function studies, has the potential to decipher in detail the molecular machinery behind a determined biological process. However, these molecular techniques have not been widely applied yet to biocontrol research, with only few examples reported in the literature. NGS techniques can have a remarkable impact on advancing knowledge on biocontrol systems through the generation of whole genome sequencing and transcriptomics (RNA sequencing, RNAseq) data allowing comparative genome and gene expression analyses to identify molecular pathways and key genes potentially involved in antagonistic activity of a given BCA. In particular, transcriptomics was used to understand how a host responds to the BCA and whether the BCA is able to induce resistance against the pathogens. For example, Zhang et al. found that *Yarrowia lipolytica* induced host resistance against *P. expansum* in apple fruit tissue through crosstalk between salicylic acid and ethylene/jasmonate pathways[10]; Yang et al. found that *Hannaella sinensis* induced genes involved in plant-pathogen interaction, plant hormone signal transduction, phenylpropanoid biosynthesis, tyrosine metabolism, glycerolipid metabolism, and α-linoleic acid pathways related to resistance regulation[11]; Zhang et al. found that *Pichia caribbica* in cherry tomatoes triggered mitogen-activated protein kinases, plant hormone signaling pathways, reactive oxygen species scavenging, and stimulated the synthesis of phenols, lignin and flavonoids, thus enhancing the resistance of cherry tomatoes against black spot[12]. Moreover, there are also examples of simultaneous RNAseq analysis of the BCA and/or the pathogen and/or the host during their interactions: Hershkovitz et al. used RNAseq to study the dual interactions of *Metschnikowia fructicola* with the mycelium of *Penicillium digitatum* and with grapefruit peel tissues, with the overexpression of genes involved in multidrug transport and amino acid metabolism by the BCA during its interaction with the fungus, and genes involved in oxidative stress, iron and zinc homeostasis, and lipid metabolism during its interaction with the host[13]; Rueda-Mejia et al. performed dual RNA-seq of *Aureobasidium pullulans* during co-incubation with *F. oxysporum*, and found that *A. pullulans* activates genes encoding secreted hydrolases and genes encoding enzymes predicted to be involved in the synthesis of secondary metabolites, while in *F. oxysporum* only 80 genes were differentially expressed, with lipid and carbohydrate metabolism being the most represented Gene Ontology categories[14]. Last, there is also an example of a three-way RNAseq during interaction of the BCA *Pseudozyma flocculosa* with *Blumeria graminis* f.sp. *hordei* on the host *Hordeum vulgare*, which allowed the elucidation of a novel mechanisms named hyperbiotrophy[15]. In particular, the authors found that *P. flocculosa* indirectly parasitizes barley in a transient manner through the activity of effectors that obtain nutrients extracted by *B. graminis* from barley leaves. The activity of these *P. flocculosa* effectors is synchronized with the activity of *B. graminis* haustorial effectors and a decline of the photosynthetic machinery of barley.

In the present study, we applied a three-way transcriptomic analysis through RNAseq to unveil the molecular mechanisms underlying the tritrophic interaction between the yeast BCA *P. terrestris* PT22AV, the fungal pathogen *P. expansum* strain 7015, and the host apple fruit *M. domestica* cv Golden delicious during their dual and tritrophic interactions.

## Results

To study at a molecular level the mechanisms underlying the tritrophic interaction between the biocontrol agent (BCA) *P. terrestris* strain PT22AV, the postharvest pathogen *P. expansum* strain 7015, and the host *Malus domestica* cultivar Golden Delicious, a transcriptomic approach based on RNAseq analysis was undertaken.

As a first step, several biocontrol assays were carried out to determine the correct conditions (timing and concentrations of the BCA and the fungus) for collecting samples for RNA extraction and analysis. Because in antagonistic assays usually one organism (either the BCA or the fungus) rapidly outcompetes the other hindering its development, the rationale of these preliminary experiments was to identify the conditions that allow the concurrent presence of the BCA *P. terrestris* and the phytopathogen *P. expansum* during their active interaction in vivo within artificial apple wounds. To this aim, different cellular concentrations of *P. terrestris* and *P. expansum* were inoculated within apple wounds, and the conidial germination was monitored with a microscope during the first 3 days of incubation. The co-inoculation of 300,000 BCA cells (inoculating 30 μL of a $1 \times 10^7$ CFU/mL yeast cell suspension) and 300 fungal conidia (inoculating 15 μL of a $2 \times 10^4$ CFU/mL conidial suspension) and 36 h of incubation appeared to be the optimal conditions to collect samples for RNAseq analysis. Although this cellular concentration of the BCA did not result in the complete protection from *P. expansum* infections, in this condition both organisms were found to be actively growing and competing in artificial apple wounds. Supplementary Fig. 1 reports microphotographs of *P. expansum* inoculated alone and with *P. terrestris* at 18, 24 and 36 h post inoculation (hpi). At 18 hpi *P. expansum* conidia started germinating both in the presence and in the absence of the BCA *P. terrestris* PT22AV (Supplementary Fig. 1a, b). At 24 hpi, in the absence of the BCA there was a high number of *P. expansum* hyphae invading the apple tissues (Supplementary Fig. 1c), while in its presence *P. expansum* hyphal growth was slightly reduced (note a non-branched hypha in Supplementary Fig. 1d). At 36 hpi branched *P. expansum* hyphae invading the apple tissues were detected both

in the absence (Supplementary Fig. 1e) and in the presence (Supplementary Fig. 1f) of the BCA *P. terrestris*. This defined 36 hpi as the most appropriate condition to collect all samples for RNAseq analysis. Control conditions were the two microorganisms incubated in rich media for 36 h, and uninoculated wounded apples.

For all the conditions three biological replicates were predisposed, with the exception of uninoculated apple and *P. terrestris* grown in vitro that consisted of two biological replicates. A total of 16 RNA samples were collected (Supplementary Fig. 2). Reads were trimmed and mapped against the reference genomes of the BCA, the fungal pathogen, and the host using STAR (see Methods for details), and the results are displayed in Supplementary Table 1. The number of mapped RNA reads against the respective reference genomes in the dual (i.e., *P. terrestris* in wounded apple tissues, or *P. expansum* in wounded apple tissue) and tritrophic (i.e., *P. terrestris* and *P. expansum* in wounded apple tissue) interactions reflected the amount of each organism present in the samples that were used for RNA extraction; as expected, this value was always higher for the apple, and it is approximatively 10% for both the biocontrol agent and the fungal pathogen in the dual interactions, and ~15% for the BCA and ~2% for the pathogen in the tritrophic interaction. Only for the sample *P. expansum* grown in vitro (replicate #1) we found that only 50% of the reads mapped against the genome.

After the mapping step, FeatureCounts was used to quantify the mapped reads in order to obtain the gene expression values for each sample. Before statistical analysis, in order to identify differentially expressed genes (DEGs), lowly expressed genes were removed with HTSFilter package and the Trimmed Means of M-values (TMM) normalization method; this package implements a novel data-based filtering procedure based on the calculation of a similarity index among biological replicates for read counts arising from replicated transcriptome sequencing data, and it maximizes the similarity between replicates, which is particularly useful in the case of mixed samples[16]. Subsequently, the quality of the processed samples was determined with the Principal Component Analysis, and statistical analysis to identify DEGs was carried out using the software EdgeR.

Because the aim of our analysis was to determine changes in gene expression of the BCA *P. terrestris* PT22AV, the fungal pathogen *P. expansum*, and of host tissues (apple) during both dual and tritrophic interactions during postharvest storage, the following seven comparisons were initially carried out:

1. *P. terrestris* in apple vs *P. terrestris* in vitro
2. *P. terrestris* + *P. expansum* in apple vs *P. terrestris* in vitro
3. *P. expansum* in apple vs *P. expansum* in vitro
4. *P. terrestris* + *P. expansum* in apple vs *P. expansum* in vitro
5. Apple + *P. expansum* vs uninoculated apple
6. Apple + *P. terrestris* vs uninoculated apple
7. Apple + *P. terrestris* + *P. expansum* vs uninoculated apple

During the initial analyses we noticed a high number of DEGs shared between the dual and tritrophic interactions, and therefore other datasets reported as "common" were generated and analyzed independently. In particular, for analysis of *P. terrestris*, DEGs with FDR < 0.05 and $\log_2$FC ± 1 obtained from the comparisons 1) and 2) were further compared to each other to identify *P. terrestris* DEGs in common to both analyzed datasets (common dataset), resulting in 3 datasets for both upregulated and downregulated genes. A further filtering was applied to select DEGs with $\log_2$FC ± 2. The same pipeline was used for *P. expansum* analysis on the comparisons 3) and 4). Last, the function of the DEGs identified was assigned through comparisons with the model organism *S. cerevisiae* for *P. terrestris* DEGs (in case of ortholog genes, *S. cerevisiae* gene nomenclature was used for *P. terrestris*), and with the sequenced *P. expansum* CMP1 isolate[17] for *P. expansum* 7015. DEGs were further subjected to gene ontology implemented with InterPRO scan and KEGG analyses as described in detail in Materials and Methods. An illustration of the main steps commonly used to manipulate *P. terrestris* and *P. expansum* datasets is reported in Supplementary Fig. 3. For analysis of the host *M. domestica*, the pipeline used to obtain DEGs is described in the relative section below.

## Analysis of gene expression in the biocontrol agent *P. terrestris* strain PT22AV

Principal component analysis revealed three distinct groups according to the experimental design (Fig. 1a). Overall, there were more upregulated DEGs of the BCA compared to the downregulated ones, with the highest number of DEGs found in the common condition between dual and tritrophic interactions.

Focusing on upregulated genes of *P. terrestris* PT22AV, were obtained 1) 135 DEGs of *P. terrestris* PT22AV when inoculated alone in artificial apple wounds; 2) 43 DEGs of *P. terrestris* PT22AV inoculated in artificial apple wounds in the presence of *P. expansum*; and 3) 624 DEGs of *P. terrestris* PT22AV commonly upregulated in the two aforementioned conditions (Fig. 1b). The complete list of *P. terrestris* PT22AV upregulated DEGs is available in Supplementary Data 1.

The 135 upregulated DEGs represent genes expressed by *P. terrestris* in vivo in apple wounds without the pathogen. GO analysis revealed that the most represented and enriched biological process categories were transmembrane transport, carbohydrate metabolic process and transport, followed by metabolic processes and branched-chain amino acid catabolic process; oxidation–reduction process included a high number of genes, although it was characterized by a low enrichment score (Fig. 1c; Supplementary Data 2). In Table 1 the most upregulated genes of this group are reported. The transmembrane transport group includes the most upregulated DEG ($\log_2$FC 7.79) *PHO89*, encoding a symporter involved in phosphate ion transmembrane transport, followed by several other plasma membrane transporter-encoding genes that mediate homeostasis and transport of glycerol, carbohydrates, allantoate, nicotinic acid, iron, and zinc. The oxidation-reduction process group includes the mitochondrial phosphatidylglycerol phospholipase C (Pgc1) involved in glycerophospholipid catabolism and the Fe(II)-dependent sulfonate/alpha-ketoglutarate dioxygenase Jlp1 involved in sulfonate catabolism for use as a sulfur source. The carbohydrate metabolic process group includes an uncharacterized glycoside hydrolase predicted to be secreted, and the transaldolase b Tal1 involved in the non-oxidative pentose phosphate pathway. Other highly expressed DEGs were not classified according to the GO and encode a MARVEL domain-containing protein generally involved in biogenesis of vesicular transport carriers, a dienelactone hydrolase, and a proteasome regulatory subunit protein. This group includes only one transcription factor that is encoded by g7108 and it is annotated as transcription factor RfeD with an unknown function. Last, KEGG analysis revealed enrichment of the propanoate metabolism, amino acids (valine, leucine and isoleucine) degradation, sulfur metabolism and biosynthesis of antibiotics (Fig. 1d; Supplementary Data 3). As a note, biosynthesis of antibiotics is a general term automatically assigned by the KEGG program, but the genes included in this group are not involved in antibiotics biosynthesis.

The common group includes 624 *P. terrestris* DEGs, which represent the genes activated by the BCA in apple wounds, regardless of the presence of the pathogen (i.e., both in the presence and in the absence of the pathogen), suggesting that these genes are important for the wound competence of the BCA. In this group, the most represented and enriched GO biological process category is transmembrane transport, which includes also highly enriched groups of polyamines, ammonium, organic cation and nucleobase transporters; oxidation-reduction process is highly represented but it has a low enrichment score (Fig. 1c; Supplementary Data 2). KEGG analysis was carried out to gain further insights into the function and the biological relevance of genes involved in this large dataset, revealing enrichment of "phenylalanine metabolism", "arginine and proline metabolism", "glycine serine and threonine metabolism", "aminobenzoate degradation" and "styrene degradation" (Fig. 1d; Supplementary Data 3). Notably, a large number of DEGs in this group encodes proteins of unknown function. The most upregulated genes of this group are reported in the Supplementary Data 4. The two most upregulated genes (g3033 and g4389) are uncharacterized, and further analyses reveal that g3033 has a NAD(P)-binding Rossmann-fold domain, while g4389 has several transmembrane domains. Transmembrane transport is the most significant

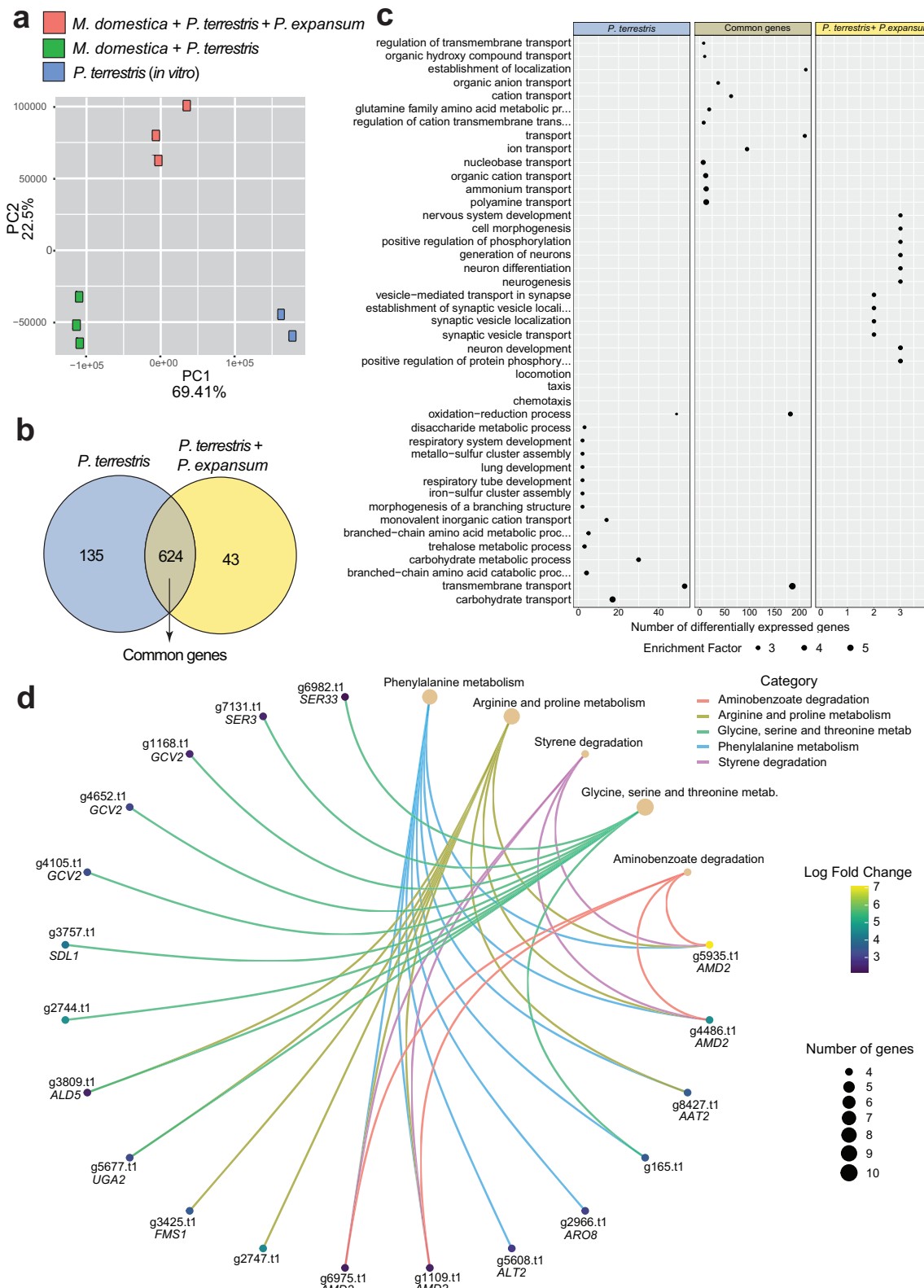

**Fig. 1 | Transcriptome analysis of the BCA *P. terrestris* during dual and tritrophic interaction with the host *M. domestica* in the absence and in the presence of fungus *P. expansum*. a** Principal component analysis of the samples subjected to RNA analysis. **b** Venn diagram showing common and unique sets of upregulated genes related to *P. terrestris* analyses; in blue are the upregulated *P. terrestris* DEGs during dual interaction with the host *M. domestica*, while in yellow are the upregulated *P. terrestris* DEGs during tritrophic interaction with the fungus *P. expansum* and with the host *M. domestica*; in light brown DEGs in common between the two aforementioned conditions. **c** Gene ontology (GO) enrichment analysis of the *P. terrestris* DEGs carried out according to three groups depicted in the Venn Diagram reported in (**b**). **d** KEGG analysis of *P. terrestris* DEGs of the common group (624 DEGs).

**Table 1 | DEGs of *P. terrestris* highly expressed (Log$_2$FC > 4) during interaction with the host**

| *P. terrestris* gene | *Saccharomyces cerevisiae* gene name | Log$_2$FC *P. terrestris* in apple | GO Description and/or SGD-NCBI-manual annotation |
|---|---|---|---|
| Transmembrane transport | | | |
| g5076.t1 | PHO89 | 7.79 | Plasma membrane Na+/Pi cotransporter |
| g1997.t1 | SEO1 | 6.90 | MFS permease member of the allantoate transporter subfamily |
| g379.t1 | TNA1 | 6.57 | nicotinic acid plasma membrane permease |
| g1017.t1 | SFC1 | 5.24 | Mitochondrial succinate-fumarate transporter |
| g6408.t1 | FRE3 | 5.15 | Ferric reductase |
| g7493.t1 | STL1 | 4.98 | Glycerol proton symporter of the plasma membrane |
| g4211.t1 | FET3 | 4.69 | Ferro-O2-oxidoreductase; multicopper oxidase |
| g4787.t1 | N/A | 4.58 | Zinc transporter |
| g6245.t1 | DAL5 | 4.32 | MFS allantoate and ureidosuccinate permease |
| Oxidation-reduction process | | | |
| g3163.t1 | PGC1 | 6.53 | Phosphatidyl Glycerol phospholipase C |
| g8163.t1 | JLP1 | 5.66 | Fe(II)-dependent sulfonate/alpha-ketoglutarate dioxygenase |
| Carbohydrate metabolic process | | | |
| g4427.t1 | N/A | 7.33 | Glycoside hydrolase with a predicted signal peptide |
| g6316.t1 | TAL1 | 5.97 | Transaldolase b of the non-oxidative pentose phosphate pathway |
| Not classified according to GO | | | |
| g3018.t1 | N/A | 5.21 | MARVEL domain-containing protein |
| g4598.t1 | N/A | 4.70 | Dienelactone hydrolase |
| g6360.t1 | N/A | 4.39 | Proteasome regulatory subunit |

Data have been extracted from Supplementary Data 1 and 2. For column 2, the gene name of the first *S. cerevisiae* hit following BLASTp has been used. DEGs were organized for GO groups and/or their function, and sorted for Log$_2$FC. Hypothetical proteins with no detected domains have not been included in this Table and can be found in Supplementary Data 1 and 2.

enriched GO biological process with 186 genes. These genes include several oligopeptide and peptide transporters orthologs of *S. cerevisiae* Opt1, Opt2, and Ptr2, several general amino acid permeases, Thi73 proposed to be involved in carboxylic acid uptake, the maltose permease Mal31, Mal11, the uridine permease Fui1, a nitrite transporter, the low glucose sensor Snf3, several transporters of urea and polyamine (Dur3, Tpo3, Tpo5, Agp2), several ammonium transporters (Amf1 and Mep2), the myo-inositol transporter Itr2, the copper oxidase Fet5, Flr1 involved in drugs efflux, Ste6 that is required for the export of pheromone factor, and several other general substrate transporters.

Moreover, a correlation between the expression and the function of the transporters reveals that the majority of them are involved in transport and uptake of organic and inorganic nitrogen, with the most represented being amino acids transporters, followed by peptides, polyamine, urea, allantoin, and the inorganic nitrogen sources nitrite and ammonium (Fig. 2). Carbohydrates and organic acids transporters were also present in this group, together with a number of general or uncharacterized transporters (Fig. 2). During BLAST analyses of the *P. terrestris* upregulated transporter-encoding genes against *S. cerevisiae*, we noticed that almost all predicted amino acid transporters were not conserved in *P. terrestris*. To clarify this, known *S. cerevisiae* transporters of amino acids[18] were subjected to reciprocal BLASTp against the *P. terrestris* genome to confirm whether they were predicted to have a conserved role. While vacuolar and mitochondrial amino acid transporter-encoding genes were conserved in *S. cerevisiae* and *P. terrestris*, the majority of the plasma membrane Yeast amino Acid Transporters (YAT) were highly divergent, and only three of them (Agp2, Dip5, Gap1) could be identified out of the 33 predicted by the genome annotation (Supplementary Data 5). Therefore, to better characterize the *P. terrestris* permeases, the 33 predicted amino acid transporters were compared to those of *S. cerevisiae* through phylogenetic analysis. The resulting tree shows three distinct phylogenetic groups, one that includes non-upregulated *P. terrestris* genes (with the exception of g5716), one that includes the *P. terrestris* transporter encoded by g4538 nested between the

majority of *S. cerevisiae* YAT, and another group that includes the majority of *P. terrestris* amino acid transporters, both DEGs and non-DEGs of the transcriptomic dataset, and the *S. cerevisiae* permease Dip5, Put4, and Agp2. These findings indicate that, with the exception of g4538, the *P. terrestris* amino acid permeases evolved compared to *S. cerevisiae* and this could likely reflect differences in their function (Supplementary Fig. 4).

The GO group oxidation-reduction process includes 182 DEGs that are predicted to play a role in oxidative stress response (Supplementary Data 2). Highly expressed DEGs within this GO group include g3330 that encodes for a nucleoside di-phosphate epimerase, g6692 encoding for NADP$^+$-dependent serine dehydrogenase and carbonyl reductase, several uncharacterized short chain dehydrogenases and oxidoreductases, the glyoxylate reductase Gor1 and Isocitrate lyase Icl1 of the glyoxylate cycle, the vacuolar protein Env9, the amidase Amd2, g7494 encoding a nitrite reductase, g4212 encoding a putative zinc-binding oxidoreductase ToxD predicted to be involved in copper metabolism, the exo-1,3-β-glucanase Exg1, the kinase Kin3, the 3-hydroxyacyl-CoA dehydrogenase and enoyl-CoA hydratase Fox2 involved in peroxisomal fatty acid β-oxidation pathway, Ser3 involved in serine and glycine biosynthesis, Lys9 involved in the lysine biosynthesic pathway.

Highly expressed genes belonging to glutamine family amino acid metabolic process include Arg4 involved in arginine biosynthesis, the succinate dehydrogenase Uga2 involved in the utilization of γ-aminobutyrate (GABA) as a nitrogen source, Asp3-4 involved in asparagine catabolism, and non-characterized glutamine synthetase, glycoside hydrolase, proline dehydrogenase, and a formamide deformylase. The GO group regulation of transcription by RNA polymerase II includes only one highly upregulated transcription factor, ortholog of Yrm1 (encoded by g7286), encoding a fungal-specific zinc finger protein that activates genes involved in multidrug resistance (Supplementary Data 2).

The last dataset included 43 upregulated DEGs expressed by *P. terrestris* in vivo in the presence of the pathogen *P. expansum*. GO analysis identified genes involved in chemotaxis, protein phophorilation, and vesicle

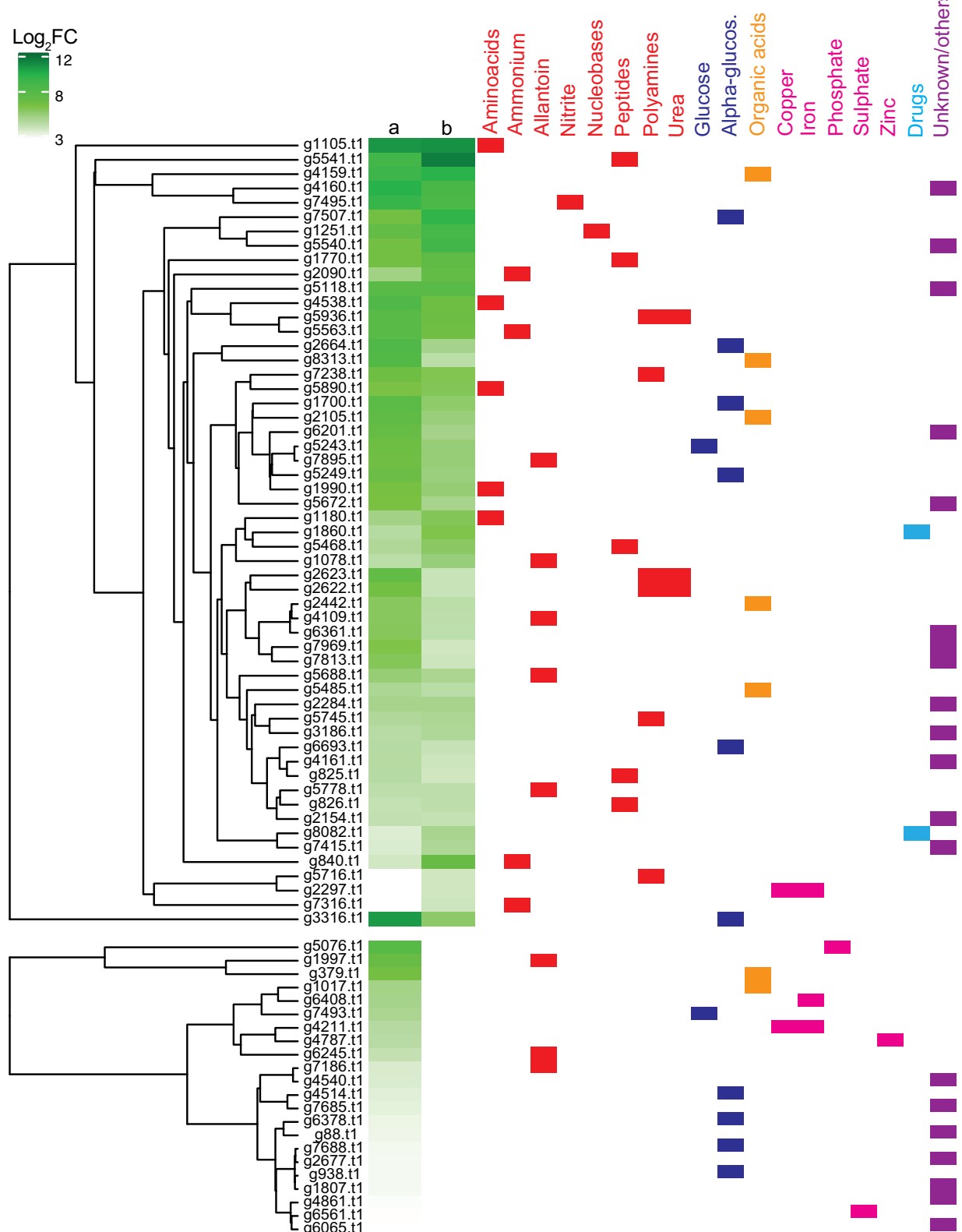

**Fig. 2 | Expression and function of the *P. terrestris* DEGs-encoding for transporters.** On the left are represented heatmap and clustering analyses based on the gene expression changes, reported as log₂FC, of the DEGs-encoding transporters of *P. terrestris* during dual interaction with the host *M. domestica* (**a**), and during tritrophic interaction with the fungus *P. expansum* and host *M. domestica*. **b** On the right it is represented the predicted function of the *P. terrestris* DEGs encoding transporters of nitrogen sources (red), sugars (blue), organic acids (orange), microelements (fuchsia), and drugs (light blue); in purple are represented transporters with unknown function or underrepresented group (e.g., only one DEG). The predicted function of the DEGs encoding transporters is based on comparison with *S. cerevisiae* orthologs where possible, with the existing information of other fungal orthologs, and with the *P. terrestris* genome annotation available.

transport, all including the same 4 genes: a more expressed ($\log_2$FC > 4) kinesin-related motor protein involved in mitotic spindle positioning *KIP2*, *KIN3* encoding a kinase involved in DNA damage response and chromosome segregation, a ras guanine nucleotide exchange factor B-like, and a unknown protein (Fig. 1c; Supplementary Data 2). In Table 2 the most upregulated genes of this group are reported. The most expressed genes were g3655 and g2837, both uncharacterized: g3655 encodes for a hypothetical protein with a N-terminal signal peptide, and it is found also in other basidiomycetes; g2837 instead has a HYalin Repeat domain, and it seems to be specific for *Papiliotrema* because GenBank search finds an ortholog only in the closely related species *P. flavescens*. Other highly expressed genes encode an uncharacterized glutathione S-transferase that is not present in the model yeast *S. cerevisiae*, a predicted F-box domain-containing protein, the UDP-glucose epimerase Gal10 involved in galactose catabolism, and a protein with a kinesin domain related to Kip2 involved in mitotic spindle positioning. Of note, also this group included a large number of genes encoding unknown proteins. KEGG analysis for this group did not provide significant results due to the low number of genes.

As regards downregulated genes of *P. terrestris*, 50 DEGs were downregulated when inoculated in artificial apple wounds alone, with the GO group inorganic anion transporter being the most enriched with four genes. Furthermore, 16 DEGs were downregulated by *P. terrestris* in artificial apple wounds only in the presence of *P. expansum*, with GO analysis displaying no enrichment categories due to the low number of genes. Last, 189 DEGs were commonly downregulated by *P. terrestris* in the two aforementioned conditions, with GO analysis revealing that response to abiotic stimulus and protein targeting are the most enriched categories. The complete lists of *P. terrestris* downregulated genes are available as Supplementary Data 6. The GO analysis of all DEGs of *P. terrestris* is reported in Supplementary Fig. 5.

Last, we searched our datasets for genes that were overexpressed in the condition "*P. terrestris* alone on apple vs *P. terrestris* in vitro" and downregulated in the condition "*P. terrestris* + *P. expansum* on apple vs *P. terrestris* in vitro", and viceversa (Supplementary Data 7). Considering a cutoff of $\log_2$FC ± 2, in the first comparison no genes were found, while in the second comparison we found 2 genes, g7404 that is absent in *S. cerevisiae* and annotated as a delayed-type hypersensitivity antigen, and g3369 that encodes a nucleoside-diphosphate-sugar epimerase whose *S. cerevisiae* ortholog is involved in pleiotropic drug resistance.

### RT-qPCR expression of predicted *P. terrestris* genes encoding nutrient transporters

To validate the results of the RNAseq analysis, the most expressed 10 *P. terrestris* genes predicted to encode nutrient transporters (*OPT1, PUT4, THI73, MAL31, FUI1, HOL1, PTR2, GAP1, MEP2,* and *OPT2*) were subjected to real time PCR during growth of the BCA in vivo in apple wounds and in vitro in apple medium. Data from in vivo samples confirm overexpression of all tested transporters at all time points tested, with the exception of *THI73* and *MAL31* at 72 hpi, and *FUI1* at 12 and 24 hpi; compared to the data from RNAseq analysis (36 hpi), qPCR showed higher expression of *OPT1, GAP1, MEP2* and *OPT2,* and lower expression for the remaining genes (Supplementary Fig. 6a). However, in the in vivo samples we could not exclude the interference of the host cDNA with the expression of the *P. terrestris* selected genes (Supplementary Fig. 6b, c), probably due to the high quantity of host cDNA present in qPCR reactions and/or to the

**Table 2 | DEGs of *P. terrestris* highly expressed (Log$_2$FC > 4) during interaction with the host and *P. expansum***

| Gene | SGD Gene name | log$_2$FC | GO Description and/or SGD-NCBI-manual annotation |
|---|---|---|---|
| Negative regulation of hydrolase activity | | | |
| g3655.t1 | N/A | 6.84 | N-terminal signal peptide |
| Galactose catabolic process | | | |
| g5316.t1 | GAL10 | 4.14 | UDP-glucose-4-epimerase |
| Microtubule polymerization | | | |
| g8073 | KIP2 | 4.02 | Kinesin-related motor protein involved in mitotic spindle positioning |
| Unclassified GO | | | |
| g2837.t1 | N/A | 5.00 | HYR domain |
| g142.t1 | N/A | 4.91 | Glutathione S-transferase |
| g1861.t1 | N/A | 4.33 | Predicted F-box domain |

Data have been extracted from Supplementary Data 1 and 2. DEGs were organized for GO groups and/or their function, and sorted for Log$_2$FC. For column 2, the gene name of the first *S. cerevisiae* hit following BLASTp has been used. Hypothetical proteins with no detected domains have not been included in this Table and can be found in Supplementary Data 1 and 2.

**Table 3 | DEGs of *P. expansum* highly expressed (Log$_2$FC > 4) during interaction with the host**

| *P. expansum* gene | Log$_2$FC | GO Description and/or SGD-NCBI-manual annotation | Studies where the gene has been identified |
|---|---|---|---|
| Oxidation reduction process | | | |
| PEX1_005170 | 5.09 | FAD-linked oxidoreductase/patulin biosynthesis cluster protein O (PatO) | |
| Transmembrane transport | | | |
| PEX1_003930 | 5.26 | Amino acid transporter *HNM1* ortholog | Ballester et al.[17] (72 h) |
| PEX1_038390 | 4.68 | Pleiotropic drug resistance protein, ABC superfamily | Ballester et al.[17] (72 h) |
| PEX1_005150 | 4.31 | CDR ABC transporter / Patulin biosynthesis cluster protein M (PatM) | |
| Fatty acid biosynthetic process | | | |
| PEX1_045200 | 4.55 | Bifunctional fatty acid transporter and acyl-CoA synthetase | |
| PEX1_002430 | 4.02 | Acyl transferase/acyl hydrolase/lysophospholipase /Non reducing polyketide synthase, patulin biosynthesis cluster protein K (PatK) | Ballester et al.[17] (72 h) |
| Unclassified GO | | | |
| PEX1_003950 | 6.89 | Hypothetical protein | |
| PEX1_050910 | 6.14 | HMG high mobility group box | |
| PEX1_003140 | 4.99 | IgE-binding protein with signal peptide | |
| PEX1_037860 | 4.02 | Cytochrome P450, E-class, group I | |

Data have been extracted from Supplementary Data 8 and 9. DEGs were organized for GO groups and/or their function, and sorted for Log$_2$FC.

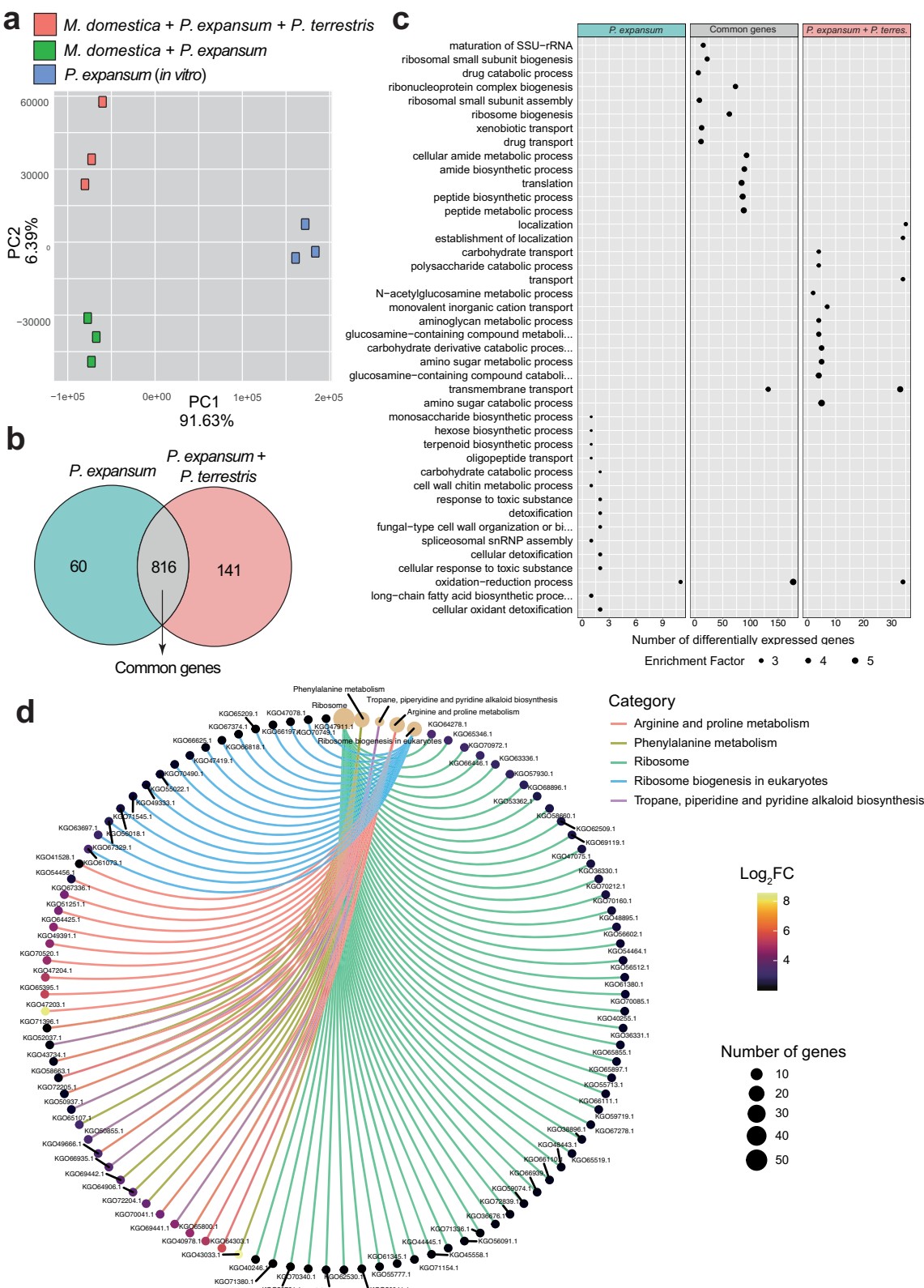

**Fig. 3 | Transcriptome analysis of the fungus *P. expansum* during dual and tritrophic interaction with the host *M. domestica* in the absence and in the presence of the BCA *P. terrestris*. a** Principal component analysis of the samples subjected to RNA analysis. **b** Venn diagram showing common and unique sets of upregulated genes between the two aforementioned conditions; in green are the upregulated *P. expansum* DEGs during dual interaction with the host *M. domestica*, while in pink are the upregulated *P. expansum* DEGs during tritrophic interaction with the BCA *P. terrestris* and with the host *M. domestica*; in grey DEGs in common. **c** GO enrichment analysis of the *P. terrestris* DEGs carried out according to three groups depicted in the Venn Diagram reported in (**b**). **d** KEGG analysis of *P. expansum* DEGs of the common group (816 DEGs).

**Table 4 | DEGs of *P. expansum* highly expressed (Log₂FC > 4) during interaction with the host and *P. terrestris***

| *P. expansum* gene | Log₂FC | GO Description and/or SGD-NCBI-manual annotation |
|---|---|---|
| Transmembrane transport | | |
| PEX1_023120 | 6.51 | Amino acid transporter |
| PEX1_010930 | 6.28 | MFS general substrate transporter |
| PEX1_072210 | 4.28 | Aquaglyceroporin like protein |
| PEX1_004560 | 4.10 | Tryptophan synthase beta chain |
| PEX1_063240 | 4.08 | MFS monocarboxylate transporter |
| Amino sugar catabolic process | | |
| PEX1_022400 | 6.63 | Exo-β-D-glucosaminidase |
| PEX1_019210 | 5.30 | Glycoside hydrolase (Cts2) |
| Oxidation reduction process | | |
| PEX1_027760 | 6.55 | Endo-chitosanase C |
| PEX1_056040 | 6.19 | Endo-chitosanase B |
| PEX1_088900 | 5.52 | Threonine dehydrogenase |
| PEX1_049140 | 4.26 | NADH-dependent flavin oxidoreductase nadA |
| PEX1_010380 | 4.10 | NAD-dependent aldehyde dehydrogenase |
| Regulation of transcription from RNA polymerase II | | |
| PEX1_103690 | 4.44 | Citrinin biosynthesis transcriptional activator |
| Unclassified GO | | |
| PEX1_012350 | 6.60 | Sialidase family/signal peptide N-region |
| PEX1_012360 | 6.15 | Dihydrodipicolinate synthetase |
| PEX1_065360 | 5.69 | RmlC-like jelly roll fold protein/signal peptide N- region |
| PEX1_005750 | 4.80 | Cysteine-type peptidase |
| PEX1_044380 | 4.61 | Acetoacetate decarboxylase |
| PEX1_071780 | 4.57 | Aminotransferase, class IV |
| PEX1_093060 | 4.11 | N-acetyl-glucosamine-6-phosphate deacetylase |
| PEX1_004560 | 4.09 | Tryptophan synthase β chain |
| PEX1_038110 | 4.02 | GPI-anchored CFEM domain protein B/signal peptide N- region |

Data have been extracted from Supplementary Data 8 and 9. DEGs were organized for GO groups and/or their function, and sorted for Log₂FC.

conservation of some gene (e.g., *TDH1*). For this reason, the expression of the same 10 *P. terrestris* genes was also tested in liquid apple medium (Supplementary Fig. 6d). All tested genes were highly expressed in apple medium at all time points, with the exception of *HOL1* that was downregulated at 12 hpi, and of *PTR2* that showed high expression only at 72 hpi. Moreover, all genes tested showed higher expression from 36 hpi, with the exception of *OPT2* that showed the highest expression at 12 hpi. Compared to the expression data obtained in in vivo apple wounds treated with the BCA, data from liquid apple medium showed higher expression in all cases, with the exception of *PTR2*. Compared to data from in vivo RNAseq analysis (36 hpi), RT-qPCR in liquid apple medium revealed higher expression of all the genes tested, with the exception of *HOL1, PTR2,* and *OPT2* (Supplementary Fig. 6d). All together the qPCR data confirm upregulation of the gene encoding nutrient transporters as identified by RNAseq analysis.

## Analysis of gene expression in the postharvest pathogen *P. expansum*

Principal Component Analysis revealed three distinct groups according to the experimental design (Fig. 3a). For *P. expansum* there were more

downregulated DEGs compared to the upregulated ones, with the highest number of DEGs found in the common dataset shared by the dual and the tritrophic interaction. In particular, focusing on the upregulated genes, we found (1) 60 DEGs of *P. expansum* when inoculated in artificial apple wounds alone; (2) 141 DEGs of *P. expansum* in artificial apple wounds in the presence of *P. terrestris*; and (3) 816 DEGs of *P. expansum* commonly upregulated in the two aforementioned conditions (Fig. 3b). The complete list of *P. expansum* upregulated DEGs is available in Supplementary Data 8.

The 60 upregulated DEGs represent genes expressed by *P. expansum* without the BCA in wounded apple tissue. GO analysis revealed that the majority of these genes are involved in oxidation-reduction process, followed by GO groups including low number of genes (<4) that are involved in biosynthetic processes, catabolic processes, and cellular detoxification (Fig. 3c; Supplementary Data 9). The most upregulated DEG encodes the uncharacterized protein PEX1_003950 (log2FC = 6.89), followed by PEX1_050910 encoding a protein with a high mobility group (HMG) (Table 3). The most expressed gene of the oxidation reduction process encodes the FAD-linked oxidoreductase PatO of the patulin biosynthetic cluster, while the Transmembrane transport group includes PEX1_003930 encoding an amino acid and polyamine transporter ortholog of *HNM1*, and two ABC transmembrane transporter proteins, one of which encodes PatM of the patulin biosynthetic cluster. The 'Fatty acid biosynthetic process' group includes PEX1_045200 encoding a bifunctional fatty acid transporter, and the polyketide synthase PatK of the patulin biosynthetic cluster. KEGG analysis corroborates GO enrichment revealing that metabolic pathway of fatty acid metabolism is significantly enriched followed by pathway related to mycotoxin biosynthesis (Supplementary Data 10).

The 816 common DEGs include those important for *P. expansum* pathogenicity and host colonization/invasion. GO analysis revealed that oxidation–reduction process is the most enriched biological process with 177 DEGs, followed by peptide metabolic process and translation, amide metabolic process and ribosome biogenesis; transmembrane transport includes a high number of genes but it is not highly enriched (Fig. 3c; Supplementary Data 9). Supplementary Data 11 reported the main upregulated DEGs divided for their function, and the studies in which these genes were identified. Representative genes of the oxidation-reduction process are the cytochrome b5 reductase Cbr1 and the cytochrome P450 monooxygenase Erg11, a nitroreductase and other flavin-containing reductases that catalyse the oxygen-independent reduction of a number of compounds, CtnA involved in citrinin biosynthesis, the glucose oxidases gox2 and gox3 involved in host tissue acidification by production of gluconic acid[19], the nitrite and nitrate reductases involved in the nitrification pathway, a number of dehydrogenases, the catalase PEX1_081840 found upregulated also by Ballester et al.[17], and others (Supplementary Data 11). The transport group includes general transmembrane transporters, and transporters of compounds used by the cells as inorganic and organic nitrogen sources, such as nitrite, nitrate, ammonium, GABA, oligopeptides, allantoin, amino acids, and nucleobases. Moreover, the common dataset includes a number of genes that encode known *P. expansum* virulence factors, such as the Peptidase S28 and the Aspergillopepsin that were found to be upregulated also by Ballester et al.[17], the secreted Protease S8 tripeptidyl peptidase I that Levin et al.[20] demonstrated by functional genetics as being involved in *P. expansum* development and virulence, and the Aspergillopepsin Pep4 found upregulated also by López-Pérez et al. in *P. digitatum* during interaction with citrus fruits[21]. Furthermore, our analysis revealed upregulation of several pectin lyase and polygalacturonase -encoding genes, as well as glycoside hydrolases such as xyloglucanases, glucanases, chitinases, and xylanases that were also identified by Ballester et al.[17]. Several transcription factors were also identified, with Arg81 involved in arginine metabolism regulation and Msn4 involved in stress response being the most expressed (log₂FC > 4); another upregulated transcription factor with log₂FC < 4 was Stm1, which is required for optimal translation under nutrient stress and is involved in TOR signaling[22]. Last, the common dataset includes a large number of highly upregulated genes encoding proteins of unknown function. Of these, the ones with predicted signal peptide at the

N-terminal region have been selected as potential *P. expansum* effectors: the two adjacent genes PEX1_077000 and PEX1_077010, the latter found overexpressed also by Ballester and colleagues[17]; the PEX1_085630 protein containing a CFEM (Common in several Fungal Extracellular Membrane proteins) domain involved in *Botrytis cinerea* pathogenicity[23], and PEX1_096670 with a membrane-bound protein non-cytoplasmic domain predicted to be located outside the membrane and that was identified also by Ballester and colleagues[17] and by Sánchez-Torres and Gonzales-Candelas[24]. KEGG analysis revealed enrichment of the ribosome biosynthesis, arginine, proline and phenylalanine metabolism, and tropane, piperidine and pyridine alkaloid biosynthesis (Fig. 3d; Supplementary Data 1).

For the condition *P. expansum* + *P. terrestris* PT22AV on apple fruit, 141 upregulated DEGs were found for $log_2FC > 2$. GO analysis revealed that the most represented biological process category was transmembrane transport, followed by amino sugar metabolic process, which included the same genes as the enriched GO categories carbohydrate derivative metabolic process and glucosamine containing compound metabolic process; oxidation–reduction process had high number of genes (>30) but low enrichment score (Fig. 3c; Supplementary Data 9). In Table 4 the main upregulated DEGs divided for their function are reported. The transmembrane transport group includes five highly expressed genes, including an unspecified amino acid transporter, a general MFS transporter, an aquaglyceroporin that transports water and other small molecules (urea, glycerol, and nitrate), a tryptophan synthase, and a MFS monocarboxylate transporter. A highly expressed exo-β-D-glucosaminidase predicted to hydrolyze chitosan and chitooligosaccharides and a glycoside hydrolase ortholog of the chitinase Cts2 are in the GO groups of amino sugar catabolic process, carbohydrate derivative metabolic process, and glucosamine containing compound metabolic process. Relevant genes of the oxidation reduction process include an endo-chitosanase C and an endo-chitosanase B also predicted to hydrolyze chitosan, and other dehydrogenases and oxidoreductases. This group includes only one highly expressed ($log_2FC > 4$) transcription factor involved in citrinin biosynthesis, and other less expressed such as the fungal specific transcription factor PEX1_058010, and PEX1_057910 encoding a zinc finger cluster protein involved in the stress response. Last, among the highly expressed protein with no GO classification there is PEX1_012350 encoding an exo-α-sialidase involved in the degradation of sialyloligosaccharides, a RmlC-like jelly roll fold protein predicted to be involved in the synthesis of L-rhamnose, a peptidase, an acetoacetate decarboxylase, an aminotransferase, and others. KEGG analysis revealed enrichment of the amino sugar and nucleotide sugar metabolism and phenylalanine metabolism (Supplementary Data 10).

One hundred eighty genes were downregulated by *P. expansum* when inoculated in artificial apple wounds alone, with the GO groups transmembrane transport and ethanol metabolic process being the most enriched. Eighty-eight genes were downregulated by *P. expansum* in artificial apple wounds only in the presence of *P. terrestris*, with GO analysis revealing low enrichment of methylation and terpene, carotene, and isoprenoid catabolic process. Last, 1068 genes were commonly downregulated by *P. expansum* in the two aforementioned conditions, with GO analysis revealing that lipid biosynthetic process is the most enriched category, followed by isoprenoid biosynthetic process, response to osmotic stress and others that have the lowest enrichment factor and less than 10 genes. A manual search revealed downregulation of genes involved in iron and copper transport (orthologs of Ctr3, Fet3, Vam3, Fet5, Atm1, Enb1, Mmt1, Zrt2, Arn2, and Sit1) and metabolism (Fre1, Fre2, Fre7, and Mig1). We found downregulation of key regulators of the cell cycle, such as the kinases Mck1 that is involved in chromosome segregation, meiotic entry, genome stability, and transcriptional regulation, Prk1 that regulates the organization and function of the actin cytoskeleton, the Cdc28 that regulates mitotic and meiotic cell cycles, Hog1 that also affects G1 and G2 cell cycle progression and is a key regulator of the osmoregulatory signal transduction cascade, Cdc37 that plays a critical role in activating cyclin-dependent kinases, and Mkk2 that is involved in control of cell integrity. The complete lists of downregulated genes are available as supplementary file (Supplementary

Data 12). The GO analysis of all DEGs of *P. expansum* is reported in Supplementary Fig. 7.

Last, all the *P. expansum*-generated datasets were searched for genes with opposite expression in the condition *P. expansum* inoculated alone in apple wounds as compared to the condition in which the pathogen was co-inoculated in apple wounds with the BCA *P. terrestris* (Supplementary Data 13). Remarkably, this comparison revealed that five genes of the patulin biosynthetic cluster (patA, patF, patM, patN, and patO) were upregulated by *P. expansum* inoculated alone on apple wounds, and downregulated when the fungus was inoculated in apple wounds in the presence of the BCA *P. terrestris*. We therefore looked at the remaining genes of the patulin biosynthetic cluster and found that, with the exception of patG that was downregulated in both conditions and patL that showed low expression, all the genes of the cluster had dramatic lower expression when fungus and BCA were co-inoculated (Fig. 4a). Expression of the genes of the citrinin, andrastin, and communesin biosynthetic clusters was also examined, and only for the citrinin biosynthetic pathway the genes PEX1_010430, PEX1_010410, PEX1_010380, and PEX1_010370 were upregulated by *P. expansum* inoculated alone on apple wounds, and downregulated when fungus was co-inoculated in apple wounds in the presence of the BCA *P. terrestris* (Fig. 4b).

## Analysis of gene expression in the host fruit *Malus domestica* Golden delicious

Principal component analysis revealed four distinct groups according to the experimental design, although samples relative to *M. domestica* during interaction with *P. expansum* showed higher variation as compared to the others (Fig. 5a).

The transcriptome of *M. domestica* was analyzed using procedure and software that were different from those used for *P. terrestris* and *P. expansum*. In particular, DEGs with FDR < 0.05 and $log_2FC \pm 1$ obtained from the comparisons number 5), 6), and 7) were further compared to each other to identify *M. domestica* DEGs in common to all analyzed datasets (reported as common dataset), resulting in 4 datasets for both upregulated and downregulated genes. A further filtering was applied to select DEGs with $log_2FC \pm 2$, and analysis was carried out according to the grouping reported in Supplementary Fig. 8. The function and the scientific name of the DEGs identified were assigned through comparisons with the model organism *Arabidopsis thaliana*. Last, for functional characterization, initially GO and KEGG analyses were performed using the same bioinformatic tools as done for *P. terrestris* and *P. expansum*. Nevertheless, we found a more appropriate classification using the specific plant database Plant MetGeneMAP (http://bioinfo.bti.cornell.edu/cgi-bin/MetGenMAP/home.cgi), which was used to perform a GO analysis with extraction of GO enriched terms, and an analysis of significantly enriched pathways. An illustration of the main steps used to manipulate the *M. domestica* datasets is reported in Supplementary Fig. 8.

For *M. domestica*, the majority of upregulated and downregulated DEGs were found during interaction with the pathogen *P. expansum*. Focusing on upregulated genes we found 1) 47 DEGs of *M. domestica* during interaction with the BCA *P. terrestris*; 2) 2132 DEGs of *M. domestica* during interaction with *P. expansum*; 3) 725 DEGs commonly upregulated by *M. domestica* during interactions with the BCA *P. terrestris* and with the fungus *P. expansum*; and 4) 17 DEGs of *M. domestica* during the tritrophic interaction with both *P. terrestris* and *P. expansum*. Note that in the latter dataset the low number of genes obtained (17 DEGs) was expected and it acts as a control validating our comparative approach through Venn diagrams, because the 2159 *M. domestica* DEGs obtained from the analysis during the tritrophic interaction with both *P. terrestris* and *P. expansum* are shared with the other conditions (Fig. 5b). The complete list of *M. domestica* upregulated DEGs obtained is available in Supplementary Data 14.

For the condition "*M. domestica* + *P. terrestris* versus uninoculated *M. domestica*", GO analysis did not find any enriched term (Supplementary Data 15), while pathway analysis revealed enrichment of the giberellin inactivation pathway (Supplementary Data 16). The most expressed genes

**Fig. 4 | Expression of the *P. expansum* genes involved in the mycotoxins biosynthesis with and without the biocontrol agent.** Expression of the *P. expansum* genes involved in the biosynthesis of the mycotoxins patulin (**a**) and citrinin (**b**) during dual interaction with the host *M. domestica* (histograms in green), and during tritrophic interaction with the BCA *P. terrestris* on the host (histograms in purple).

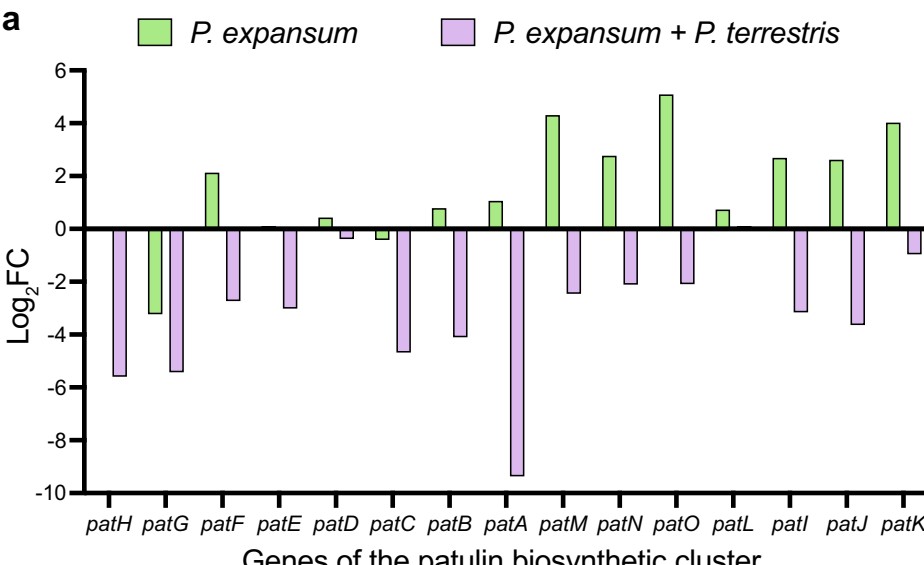

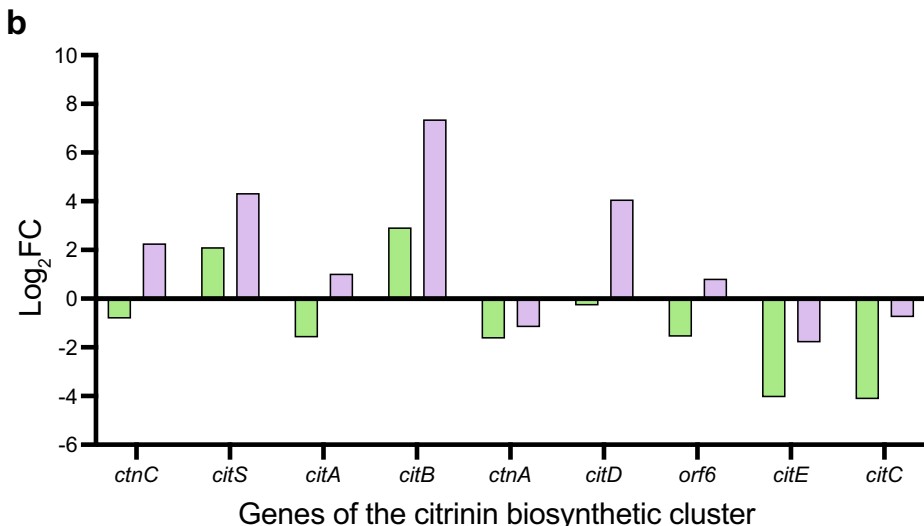

of this dataset are *POT5* involved in potassium transport, the monocopper oxidase-like protein *SKU5* predicted to be involved in directed root tip growth[25], *BGLU13* encoding a β-glucosidase that could play a role in plant defense[26], *GA2OX6* encoding a gibberellin 2-oxidase that regulates plant growth by inactivating endogenous bioactive gibberellins[27], and the transcription factor *UPB1* that regulates the expression of a set of peroxidases that modulate the balance of reactive oxygen species in cell cycle progression[28].

Analysis of the *M. domestica* DEGs during the interaction with the pathogen *P. expansum* identified a number of enriched GO terms, including post-translational protein modification and protein amino acid phosphorylation, phosphate metabolic process, response to stress, wounding, and biotic stimulus, immune response, innate immunity, cell communications, and others less enriched (Fig. 5b; Supplementary Data 15). The jasmonic acid biosynthesis was the only significantly enriched pathway (Supplementary Data 16). Accordingly, analysis of the KEGG database found enrichment of the plant pathogen interaction and MAPK signaling pathway, followed by amino acids biosynthesis, and metabolism of several lipids (glycerophospholipids, phospholipids, glycerolipids, linoleic acid) and glucosinate (Supplementary Data 16).

Analysis of the *M. domestica* common DEGs identified during the interaction with the pathogen or the BCA revealed enrichment of the GO terms response to stimulus, response to stress, and metabolic processes

(Fig. 5b; Supplementary Data 15), while phenylethanol biosynthesis is the only significantly enriched pathway (Supplementary Data 16). Last, for the 17 *M. domestica* DEGs upregulated during the tritrophic interaction with both *P. terrestris* and *P. expansum* no enriched GO terms were found (Supplementary Data 15), and pathway analysis found only the ascorbate glutathione cycle (Supplementary Data 16).

As regards downregulated genes, we found 1) 171 genes of *M. domestica* during interaction with the BCA *P. terrestris*; 2) 2399 genes of *M. domestica* during interaction with *P. expansum*; 3) 469 genes commonly downregulated by *M. domestica* during the interaction with the BCA *P. terrestris* or with the fungus *P. expansum*; and 4) 5 downregulated DEGs by *M. domestica* during the tritrophic interaction with both *P. terrestris* and *P. expansum* (Fig. 5c). The complete list of *M. domestica* downregulated DEGs obtained is available in Supplementary Data 17.

For the condition "*M. domestica* + *P. terrestris* versus uninoculated *M. domestica*", GO analysis of downregulated genes found enrichment of GO category "response to stimulus" (Fig. 5c; Supplementary Data 18), while pathway analysis found enrichment of homogalacturonan degradation, ethylene biosynthesis from methionine, glutamate degradation, flavonoid biosynthesis, and tyrosine biosynthesis (Supplementary Data 19). For the condition "*M. domestica* + *P. expansum* versus uninoculated *M. domestica*" GO analysis found a high number of terms, with the most enriched being photosynthesis, response to abiotic stimulus, response to radiation and light,

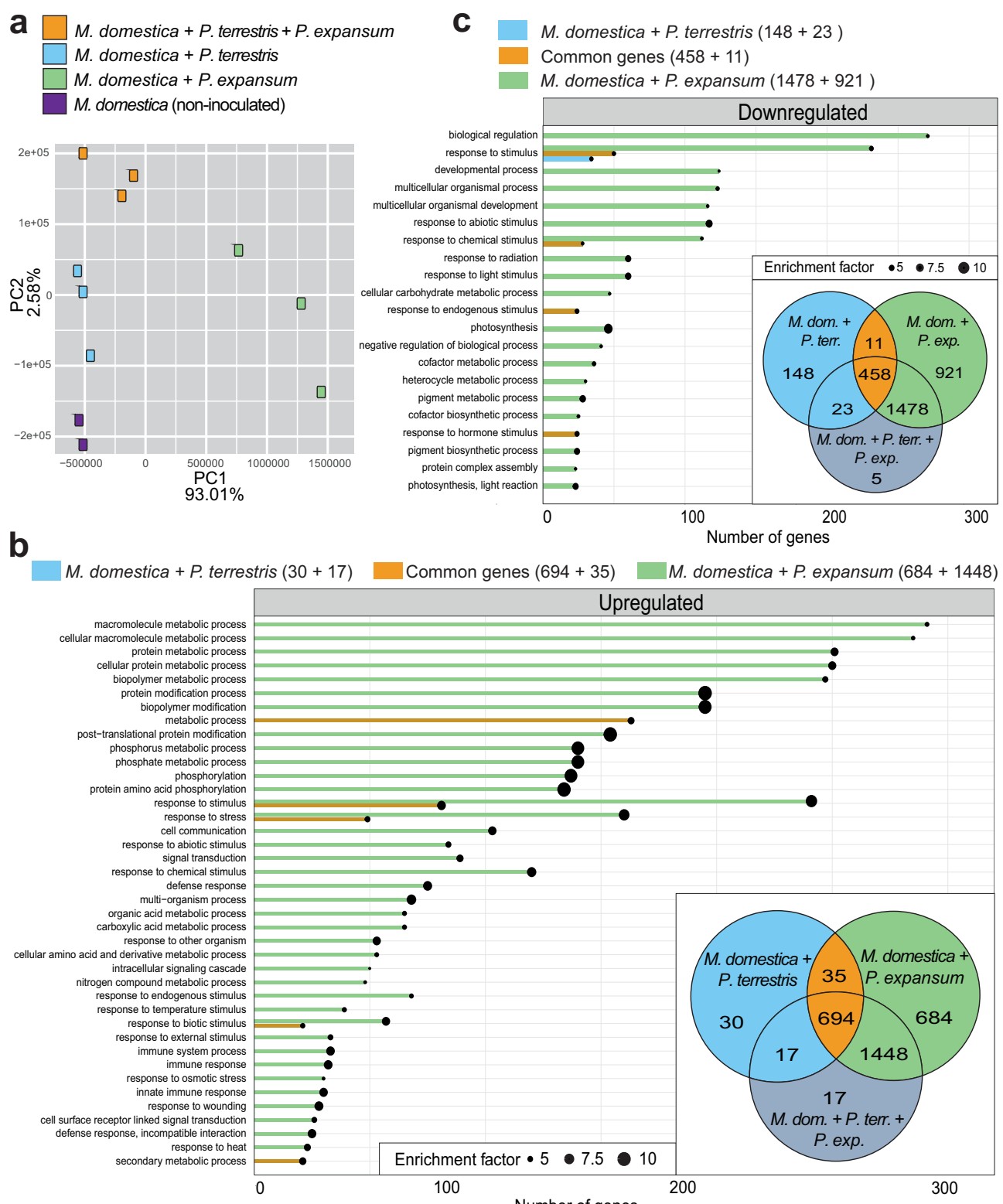

**Fig. 5 | Transcriptome analysis of the host *M. domestica* during interaction with the BCA *P. terrestris*, with the fungus *P. expansum*, and with both of them.** **a** Principal component analysis of the samples subjected to RNA analysis. **b** GO enrichment analysis of the *M. domestica* upregulated DEGs carried out according to color-coded groups depicted in the Venn Diagram reported in inset of (**b**). **c** GO enrichment analysis of the *M. domestica* downregulated DEGs carried out according to color-coded groups depicted in the Venn Diagram reported in inset of (**c**).

and pigment metabolic process (Fig. 5c; Supplementary Data 18). Pathway analysis found enrichment of photosynthetic processes and photorespiration, and biosynthesis of starch, chlorophyll, and carotenoids (Supplementary Data 19).

For *M. domestica* downregulated common DEGs identified during the interaction with the pathogen or the BCA, GO analysis revealed enrichment of terms related to response to stimulus (Fig. 5c; Supplementary Data 18), while significantly enriched pathways were carotenoid and brassinosteroid biosynthesis (Supplementary Data 19). Last, for the 5 *M. domestica* DEGs downregulated during the tritrophic interaction no enriched GO terms were found (Supplementary Data 18), and pathway analysis found only the stachyose biosynthesis (Supplementary Data 19).

Last, we aimed to understand in detail the contribution of genes involved in plant defense and plant immunity in response to the BCA *P. terrestris*, to the phytopathogenic fungus *P. expansum*, and to both of them. We focused on genes known to be involved in pathogen-associated molecular patterns (PAMPs) triggered immunity (PTI) and effector triggered immunity (ETI), along with genes encoding heat shock proteins and genes of the calmodulin, ethylene, jasmonic acid, and salicylic acid pathways, given their known role in plant immunity[29]. In general, there was a clear trend of upregulation by the host in the presence of *P. expansum* and, conversely, downregulation or low expression in the presence of *P. terrestris* (Fig. 6). As concerned PTI, the conserved receptor-like kinase *BAK1* (encoded by MD08G1221700) that serves as signaling kinase for many of the PAMP receptors, and the *CERK1* receptor kinase (encoded by MD09G1111800) known to sense chitin oligomers released from the fungal cell wall have, respectively, a $\log_2$FC = −0.23 and 0.59 in the presence of the BCA *P. terrestris*, $\log_2$FC = 2.13 and 3.24 in the presence of the *P. expansum*, and $\log_2$FC = 0.93 and 1.63 during tritrophic interaction. Likewise, the kinases *MEKK1* (encoded by MD15G1358300), *MKK1/2* (encoded by MD15G1212100), *MPK4* (encoded by MD17G1173100) and *MKS1* (encoded by MD01G1011800), which function together in a mitogen-activated protein cascade to regulate plant immunity[30] have, respectively, a $\log_2$FC = −0.40, −0.26, −0.01 and −0.48 in the presence of the BCA *P. terrestris*, $\log_2$FC = 2.03, 0.32, 1.21, and 2.33 in the presence of the *P. expansum*, and $\log_2$FC = 0.78, −0.12, 0.17 and 0.79 during tritrophic interaction. The *MKS1* kinase targeting transcription factor *WRKY33* (encoded by MD04G1167700), a key positive regulator of immunity against necrotrophic pathogens[31], has $\log_2$FC = −1.52 in the presence of the BCA *P. terrestris*, $\log_2$FC = 3.02 in the presence of the *P. expansum*, and $\log_2$FC = 0.84 during tritrophic interaction. Conversely, the kinases *MKK4/5* and *MPK3/6* that also target *WRKY33* are not differentially expressed in any of the conditions studied. With few exceptions, the majority of the WRKY transcription factors that are involved in the regulation of defense gene expression in plant immunity are downregulated during the interaction of *M. domestica* with the BCA *P. terrestris*. Furthermore, the receptor-like cytoplasmic kinase *BIK1* (encoded by MD07G1199400) that contributes to the activation of the calcium influx is also upregulated only in the presence of *P. expansum* and in the tritrophic interaction; *BIK1* also contributes to ROS production by phosphorylating the oxidase homolog D RbohD (encoded by MD13G1134500), which is also upregulated in the presence of the pathogen ($\log_2$FC = 3.74) and during the tritrophic interaction ($\log_2$FC = 2.11), whereas it is downregulated in the presence of the BCA *P. terrestris* ($\log_2$FC = −0.60). The calcium related protein kinase *CPK4*, *CPK5*, *CPK6*, and *CPK11* are also involved in promoting ROS production via RbohD; *M. domestica* has orthologs of *CPK4* and *CPK5*, both differentially upregulated only in the presence of *P. expansum* and during the tritrophic interaction. Other genes involved in calcium signaling and homeostasis, such as calmodulin-binding transcription factors *CAMTa3*, *CBP60g* and *CBP60a*, crucial regulators of plant defense, are upregulated only in the presence of *P. expansum* and during the tritrophic interaction.

As concerns ETI, the genes *EDS1* (enhanced disease susceptibility 1), *NRG1* (N requirement gene 1), senescence-associated gene 101 (*SAG101*), phytoalexin deficient 4 (*PAD4*), and *ADR1* (activated disease resistance 1) constitute major non-interchangeable signaling node in nucleotide-binding

leucine-rich repeat proteins (NLRRs)-mediated ETI[32]; in our analysis, *EDS1* (encoded by MD14G118700) and *NRG1* (encoded by MD02G1165500) are differentially upregulated only in the presence of *P. expansum, PAD4* (encoded by MD15G1136300) is differentially upregulated in all conditions, although at higher level in the presence of *P. expansum*, *ADR1* (encoded by MD08G1041800 and MD08G1042400) is downregulated in all conditions, and *SAG101* (encoded by MD15G1136300) is not differentially regulated.

Furthermore, some genes of the ethylene pathway, such as the ethylene receptor *ETR1* (encoded by MD12G1246000), the ethylene-forming enzyme *ACO4/EAT1* (encoded by MD17G1106300), and the kinases *CTR1* (MD12G1017800) are downregulated in the presence of the BCA *P. terrestris* and upregulated in the presence of *P. expansum* and during tritrophic interaction. However, several others genes (*COI, NINJA, WRK70*) of the ethylene pathway do not show a differential expression between the different treatments. Furthermore, genes of the jasmonate pathway involved both in jasmonate synthesis (*LOX, OPR3, JAR1*) and signaling (*JAZ* genes)[33] were downregulated in the presence of the BCA and upregulated in the presence of *P. expansum* and during tritrophic interaction.

Last, we assessed the expression of genes involved in salicylic acid (SA) biosynthesis. SA biosynthesis occurs via the isochorismate synthase (ICS)- and the phenylalanine ammonia-lyase (PAL)-derived pathways, which utilize chorismate as the common precursor[8]. Blast search revealed that *M. domestica* contains only one isochorismate synthase gene, and four orthologs of the *PAL1* gene. In our dataset we found high expression ($\log_2$FC > 8) of the isochorismate synthase *ICS1* (encoded by MD14G1195500) when the host fruit was inoculated with the fungus, and lower expression ($\log_2$FC < 3) when the host was inoculated with the BCA *P. terrestris* or both with the fungus and the BCA. As regards the PAL pathway, we found overexpression of three of the four *PAL1* orthologs (encoded by MD12G1116700, MD01G1106900, MD07G1172700) when the host was infected with the fungus and during tritrophic interaction, and downregulation of the other ortholog (encoded by MD04G1096200) when the host is inoculated with the BCA *P. terrestris*. SA effects occur through the activity of the transcription coactivator "nonexpressor of *PR* genes" (*NPR1*), which interacts with several TGA (TGACG-binding) transcription factors which bind to the promoter of the SA-dependent defense gene *PR1* and activate its expression[34]. *M. domestica* genome includes at least two orthologs of *NPR1* (MD10G1236700 and MD05G1256300), which are both downregulated in the presence of the BCA and upregulated in the presence of the fungal pathogen during tritrophic interaction.

## Discussion

In the pathosystem we assessed in our study, biocontrol is characterized by a tritrophic interaction that is established between the antagonist yeast *P. terrestris* strain PT22AV that operates as a BCA to protect the fruit tissues, the necrotrophic phytopathogenic fungus *P. expansum* strain 7015 that aims to kill the fruit tissues to obtain the nutrients necessary for its development, and the host fruit *M. domestica* (cv Golden Delicious) that responds to both of them. Understanding the mechanisms underlying this interaction is crucial to potentiate the biocontrol activity of the yeast BCAs. To this aim, in the present study we applied a transcriptomic approach based on RNAseq to identify genes expressed by the three organisms during their dual and tritrophic interactions. Moreover, it is important to note that we used classical growth media for both *P. terrestris* (YPD) and *P. expansum* (PDB) as comparative conditions, with the aim to highlight for both the BCA and the pathogen possible mechanisms that regulate the nutritional shift from the growth media to the apple tissues. For analysis, the initial datasets were compared to each other through Venn diagram to obtain for each organism (*P. terrestris, P. expansum* and *M. domestica*) a dataset of the genes in common or specific for the conditions under examination (Fig. 1b, Fig. 3b, and Fig. 5b, c).

For the BCA *P. terrestris*, data analysis revealed that the highest number of upregulated DEGs is included in the "common group" dataset, suggesting a significant transcriptional rewiring operated by the BCA to colonize apple wounds both in the presence and in the absence of the fungal pathogen

**Article**

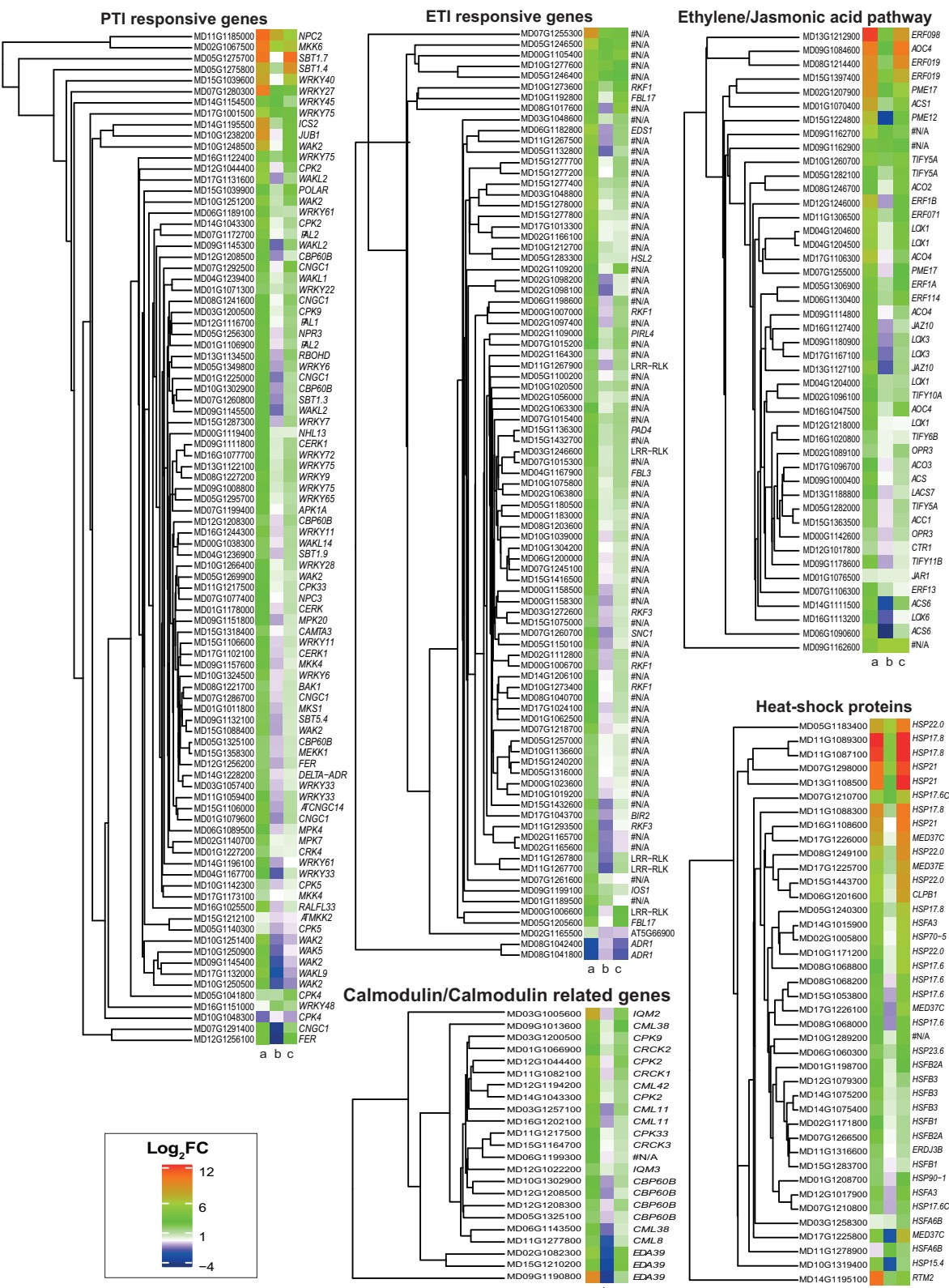

**Fig. 6 | Expression of the *M. domestica* genes involved in plant immunity and plant defense.** Heatmaps reporting the expression of the *M. domestica* genes known to be involved in plant immunity and plant defense response during interaction with the fungus *P. expansum* (**a**), with the BCA *P. terrestris* (**b**), and with both of them (**c**). Genes involved in pattern-triggered immunity (PTI), effector-triggered immunity (ETI), ethylene and jasmonic acid pathways, genes encoding heat shock proteins, and calmodulin and calmodulin-related genes were identified as described in Materials and Methods, and searched in *M. domestica* genome to identify their orthologs; the name of the *A. thaliana* ortholog is used where available.

(Fig. 1b). These common DEGs include *P. terrestris* genes important for wound competence. On the basis of the analyses carried out, these DEGs are mainly involved in nutrients transport and oxidative stress resistance (Fig. 1c). Among the transporters, the most represented are permeases involved in the uptake of both organic (amino acids, peptides, etc) and inorganic nitrogen (ammonium and nitrite) sources (Fig. 2), indicating that nitrogen is most likely the main nutrient that the BCA and the phyto-pathogenic fungus are competing for. A time course expression analysis of the most upregulated transporters confirms RNAseq data, and revealed that the majority of the transporters are highly expressed up to 3 days of incubation (Supplementary Fig. 6a, d). This suggests that the nutrients uptake operated by the BCA to outcompete the pathogen plays an important role during the entire process of BCA-pathogen interaction, and not only at the beginning of the microbial interaction as originally hypothesized[35]. Amino acids are a preferred nitrogen source for microbial growth as they can be easily assimilated, and can be also used as both nitrogen and carbon sources[36]; amino acids can also enter the cells as peptides taken up by specific permeases (Ptr2, Opt1, and Opt2)[36,37]. Accordingly, genes encoding for amino acid and peptide permeases were the most abundant and the most expressed in the *P. terrestris* dataset, regardless of the presence or absence of the fungus *P. expansum* (Fig. 2; Supplementary Data 1). Intriguingly, we noticed a correlation between the functional profile of the upregulated transporters of the BCA and the nutrient composition of the apple Golden Delicious used in this study, which includes mainly free amino acids, followed by sugars, organic acids, phenolic compounds, and fatty acids[38], suggesting that *P. terrestris* activates permeases-encoding genes to readily uptake and utilize nutrients already available in the apple tissue. Although a time course analysis of amino acids content in apple wounds untreated and treated with *P. terrestris* does not support this hypothesis (Supplementary Fig. 9), the role of amino acid transport in the biocontrol activity of *P. terrestris* against *P. expansum* cannot be excluded, and the slight amino acids increase detected as later time points might indicate that the fruit synthetizes novel amino acids to replace those metabolized by the BCA, or that these amino acids may derive from proteolytic activity by the BCA or by the apple fruit tissue itself. Overexpression of permeases predicted to transport other nitrogen sources suggest that such sources might also be components of the apple but have not been detected in previous studies, or they might derive from microbial metabolism as reported for polyamines, ammonium, nitrite, urea, and allantoin[39–41].

While BLAST searching to infer the function of the upregulated *P. terrestris* transporters, we noticed that the majority of them have *S. cerevisiae* orthologs that are regulated by the nitrogen catabolite repressor (NCR) system, and hence we evaluated whether NCR might control the response of *P. terrestris* to the apple tissues. NCR is a transcriptional mechanism that enables the downregulation of genes involved in the utilization of poor nitrogen sources when preferred ones are available[42]. In *S. cerevisiae*, NCR is regulated by four GATA family zinc finger transcription factors, the two transcriptional activators Gln3 and Gat1 that positively regulate the expression of permeases and pathway-specific catabolic enzymes upon nitrogen starvation, and the two repressors Dal80 and Gzf3 that down-regulate *GAT1* transcription under nitrogen-rich conditions. When optimal nitrogen sources are available, Gat1 is sequestered in the cytoplasm by phosphorylation mediated by the TOR kinases Tor1 and Tor2. The phos-phorylated form of Gat1 is also bound by the regulatory protein Ure2, which sequesters it in the cytoplasm thus preventing its translocation into the nucleus and the transcription of NCR-sensitive genes[42]. In agreement with previous findings in *Neurospora crassa, Aspergillus fumigatus* and in the basidiomycetes *Cryptococcus neoformans*[43], also in the *P. terrestris* genome there is only one predicted NCR activator *GAT1* (g1827) and one NCR repressor *GZF3* (g8117). Interestingly, these two key NCR genes were not differentially regulated in our dataset (Supplementary Fig. 10), suggesting that the NCR machinery is likely not activated by the BCA *P. terrestris* during host tissue colonization. This hypothesis is also supported by the growth kinetics of *P. terrestris* in a medium mimicking the apple tissue, which it clusters with the growth in the presence of favorite nitrogen sources,

such as arginine, ammonium, proline, glycine, serine, etc (Supplementary Fig. 11).

Because NCR is likely not active in *P. terrestris*, how does this yeast BCA sense and respond to the nutritional composition of the host? Besides NCR, in the model yeast *S. cerevisiae* there are other regulatory circuits to sense and utilize nitrogen and free amino acids, like the SPS sensor system, the Global Amino Acid Control (GAAC), the transceptor-mediated amino acid sensing, and the target of rapamycin (TOR) pathway[40,42]. The SPS sensor regulates the expression of genes involved in amino acid catabolism and it is composed by Ssy1, Ptr3, Ssy5, an Stp1 and Stp2[36,42]; these components were not found in the genome of *P. terrestris* (Supplementary Fig. 10), and are also absent in *C. neoformans*[44], implying a different evolutionary mechanism of amino acids sensing in the basidiomycete yeasts belonging to the Tremellomycetes. Likewise, the main activator of the GAAC pathway, the protein Gcn4, is also missing in the genome of basidiomycete yeasts (Supplementary Fig. 10). The TOR pathway is the main sensor of the intracellular nutrient state of the cells, in particular of amino acids[45]. While *S. cerevisiae* has two *TOR* genes, *TOR1* and *TOR2*, the majority of the fungi has only a *TOR1* ortholog. *Papiliotrema terrestris* genome has a single *TOR1* ortholog (g6083), and a Tor-like kinase (g4917) gene that is ortholog of the *TLK1* gene of *C. neoformans* (Supplementary Fig. 10). *Cryptococcus neoformans TOR1* is essential for viability and it functions like both *S. cerevisiae TOR1* and *TOR2*, as it is involved in cytoskeleton organization, metabolic processes, ribosome biogenesis, stress response, and signal transduction[46]. *Cryptococcus neoformans* Tlk1 is a kinase involved in stress resistance, but not in nutrient sensing[47]. Low expression of the *P. terrestris TOR1* gene indicates that the TOR pathway is not activated by the BCA during apple wound colonization, while the higher expression of *TLK1* might be related with its predicted function in stress resistance (Supplementary Fig. 10). Finally, another sensing mechanism for extracellular amino acids is provided by transceptors, which act simultaneously as transporters and receptors. The *S. cerevisiae* transceptor *GAP1* is regulated in response to intracellular amino acid abundance through oligo- and polyubiquitination-guided endocytosis, and it is a transceptor for all L-amino acids, some D-amino acids, citrulline, and polyamines[42,48,49]. While *Candida albicans* has also transceptor proteins, it is unknown whether other fungi, including the basidiomycete model *C. neoformans*, have functional transceptors. In *P. terrestris* a single permease encoded by g4538 is phylo-genetically related to the majority of the *S. cerevisiae* permeases and it is the closest ortholog of *S. cerevisiae GAP1* (Supplementary Fig. 4; Supplementary Data 5). Even though the *P. terrestris* g4538 is highly expressed in our datasets and it could be speculated that it functions as transceptor, there are other permeases that are expressed at higher levels than g4538 (Fig. 2), and therefore their potential role as transceptors can only be determined following functional genetics studies.

Besides nitrogen, transporters of sugars and organic acids were also overexpressed by the BCA during interaction with the host and the fungus, indicating that they are also part of the nutrient competition between the BCA and the phytopathogenic fungus. Five of the six upregulated sugar transporters are classified according to the GO database as general α-glucosidase permease and general transporters (Fig. 2), and they share homology with *S. cerevisiae* transporters *MAL31, MAL11, HXT2,* and *ITR2* involved in maltose, glucose, and myo-inositol transport, respectively, while g5243 is annotated as a high-affinity glucose transporter and corresponds to *S. cerevisiae* Snf3, which is a glucose, fructose and mannose sensor. Given that the three main sugars found in apple are glucose, fructose, and saccharose[38], it seems reasonable that they are uptaken by the five identified *P. terrestris* transporters, with a major role played by the most expressed g7507, encoding a transporter of the fungal hexose subfamily of the major facilitator superfamily. Therefore, also *P. terrestris* sugar transporters do not have a conserved function compared to *S. cerevisiae*. The importance of the genes involved in amino acids, peptides, and sugar transport has been also reported by Hershkovitz et al. during dual interaction of the BCA *Metsch-nikowia fructicola* with both the pathogen (*P. digitatum*) and the host (grapefruit)[13].

Resistance to oxidative stress is the other process identified as important for the BCA, specifically for wound competence (Fig. 1c). The importance of the oxidative stress resistance for antagonistic activity of *Papiliotrema* BCAs has been well documented[3,6,50]. This mechanism is strongly correlated with the competition for nutrient and space as it allows the *P. terrestris* BCA to rapidly adapt to and colonize fruit tissues through resistance to reactive oxygen species (ROS) generated in fruit after wounding[51], as we demonstrated through biochemical and genetic approaches[3,6]. Oxidative stress resistance has been shown to be important for biocontrol activity also in the ascomycetes *C. oleophila*[52] and *M. fructicola*[53]. Intriguingly, in a previous study we demonstrated that a *P. terrestris* mutant for the gene *YAP1*, a core transcription factor involved in oxidative stress resistance, displayed lower antagonistic activity against *P. expansum* and *M. fructigena*[3]. However, *YAP1* is not a DEG in our dataset, indicating in this case lack of correlation between functional genetics and gene expression, as observed also in other biological contexts[54,55].

We next focused on the other interactions that characterize the BCA to determine whether there are specific traits exclusively due to the absence or the presence of *P. expansum*. In the dual interaction of the BCA with the host, carbohydrate transport is the most enriched and represented GO process. Intriguingly, with the exception of *MAL11* and *ITR1*, in the dual interaction BCA-host there are different sugar transporters compared to the common group described before (Fig. 2), with a high-affinity glucose transporter ortholog of the glycerol proton symporter Stl1, and the α glucoside permeases Mph2 and Mph3 that are predicted to transport maltose, maltotriose, alpha-methylglucoside, and turanose[56]. The involvement of other sugar transporters in the dual interaction might reflect the lack of competition with the fungus with a consequent different nutritional requirement of the BCA. This hypothesis is also corroborated by the low number of nitrogen transporters, and by the upregulation of several transporters for micronutrients, such as phosphate, iron, zinc and sulphate, which were instead not highly represented in the common group (Fig. 2). In partial agreement with our study, during interaction of the BCA *M. fructicola* with the host only genes involved in uptake and transport of iron and zinc were upregulated, while transporters of other microelements were upregulated only during interaction with the fungus[13]. Moreover, micronutrients and iron in particular are crucial for the biocontrol activity of some BCAs such as *M. pulcherrima* and *M. fructicola*[35], while their role seems minimal or null for *P. terrestris*, indicating that biocontrol agents can operate through different mechanisms to display their antagonistic activity.

The last group of upregulated DEGs includes *P. terrestris* genes expressed in vivo only in the presence of the pathogen *P. expansum* and hence potentially involved in microbial interactions between the BCA and the fungus. However, none of the found genes seems to play this role, with functional analysis assigning the chemiotaxis GO group to the proteins Kip2 and Kin3 involved in chromosome segregation and mitotic spindle positioning, and to an uncharacterized ras guanine nucleotide exchange factor. Nevertheless, this group includes a large number of proteins with uncharacterized function and it cannot be excluded that some of them have a direct role in the microbial interaction between the BCA and the fungus. Moreover, it was also expected that this group included hydrolases-encoding genes that contribute to the antagonistic activity of *P. terrestris*, since it has been reported that a closely related BCA produces β-1,3-glucanase when incubated in the presence of *P. expansum* hyphal cell wall[4]. Surprisingly, we found high expression of the *P. terrestris* exo-1,3-β-glucanase Exg1 and of a predicted chitinase both in the presence and in the absence of the pathogen (Supplementary Data 1), suggesting that their expression is triggered by the apple tissues and not by the fungal pathogen, and therefore they are not specifically involved in fungal cell wall degradation but also in other biological processes. Similarly, in *M. fructicola* Hershkovitz et al. found that a chitinase was upregulated both in the dual interaction with the pathogen and the host, while a glucanase was specifically induced only in the presence of the pathogen[13]; analogous findings were also reported for *Pichia anomala*[57]. In a recent transcriptomic study Rueda-Mejia et al. found instead a predominant role of several hydrolases in the biocontrol activity of

*Aureobasidium pullulans* against *Fusarium oxysporum*[14]. These studies together suggest that hydrolases can be subjected to different regulatory mechanisms and their contribution to the antagonistic activity of the yeast BCAs can be diverse and should be assessed with studies of functional genetics, as done in *C. oleophila*[58,59] and *P. anomala*[60,61].

As regards the fungus *P. expansum*, it is one of the most important necrotrophic fungal pathogen especially in postharvest, and it has been intensively studied at a molecular level[62]. In our analysis we found that the highest number of DEGs is included in the "common dataset", suggesting a significant transcriptional rewiring by the fungus both in the presence and in the absence of the BCA (Fig. 3b). Upregulated DEGs of the common group include *P. expansum* genes demonstrated by functional genetics to be important for pathogenicity and host colonization/invasion, such as for example the glucose oxidase Gox2 that is involved in the conversion of glucose in D-gluconic acid, which is responsible for host tissue acidification and consequent activation of genes important for cell wall degradation and tissue maceration[19], and the secreted peptidase S8 involved also in vegetative growth, conidiation and autophagy[20]. Moreover, our analysis found a number of genes that are predicted to be involved in pathogenesis, such as gox3, and several proteases, pectinases, glycoside hydrolases, and effectors (Supplementary Data 8,9,11). Remarkably, the majority of these genes were already identified in other transcriptomics approaches[17,21,24,63], hence validating the experimental design that we used in our study. In particular, our results mainly overlap with those obtained by Ballester et al.[17] at 48 hpi (Supplementary Data 11), most likely because of the similar experimental design (in our case RNA samples for sequencing were collected after 36 hpi), and of the similarity between our *P. expansum* strain 7015 with their *P. expansum* strain PEX1, whose annotation was used in our RNAseq pipeline (see Material and Methods for details). Nevertheless, differences with the study of Ballester et al.[17] concerned mainly the DEGs encoding nutrient transporters, which might reflect the different comparative control condition used [PDB in our case, and a mix of healthy apple tissues and fungal spores in the study of Ballester et al]. The use of a different medium for the control condition in the comparison with the in vivo samples allowed to highlight also the *P. expansum* genes required for the nutritional shift, similarly to what is observed for the BCA *P. terrestris*. Indeed, also the *P. expansum* DEGs of the common group included a large number of transporters for organic and inorganic nitrogen, sugars, organic acids, and microelements that are predicted to be upregulated to uptake apple nutrients (Supplementary Fig. 12). Last, we also found high upregulation of several genes of the patulin biosynthetic cluster (Supplementary Data 8), which instead were not differentially expressed in the study of Ballester et al.[17]. This difference might be due to the intrinsic diversity in patulin production by different *P. expansum* strains, as seen by Ballester[17] and Bartholomew[64], by the experimental conditions used for analysis [liquid vs solid media[65]; or static vs shaking[66]], and by the different pathosystems, including the apple cultivars and their physiological parameters of maturity[67]. It could be speculated that elevated expression of the patulin biosynthetic genes reflects an active role of the mycotoxin in the infective process of *P. expansum* strain 7019 used in our experiment, although this hypothesis needs to be confirmed by functional genetics. Indeed, there are conflicting results on the role of patulin as a pathogenicity factor, and it seems that it can contribute to virulence in a cultivar-dependent manner[17,62,67–69].

Beside this information on *P. expansum* pathogenicity corroborated by other studies, our experimental plan provides the opportunity to unveil how the phytopathogen *P. expansum* is affected by the presence of the antagonist BCA *P. terrestris* during interaction in vivo on the host. We found enrichment of metabolic process of aminosugar and related products (Fig. 3c), which includes a highly expressed chitinase and an exo-β-D-glucosaminidase predicted to degrade fungal chitin and chitosan, respectively; as a note, *P. expansum* DEGs included also two other highly expressed enzymes that degrade chitosan, such as an endo-chitosanase B and a endo-chitosanase C (Supplementary Data 8). It can be hypothesized that these degradative enzymes produced by *P. expansum* are active on the cell wall of

the BCA *P. terrestris* and have an ecological function aiming at eliminating competitor microbes, or they might have a nutritional role[70].

Last, to further assess possible reciprocal influence of the BCA on the fungus, the *P. expansum* datasets were searched for genes that had opposite expression during dual and tritrophic interactions. Remarkably, we found that the presence of the BCA during *P. expansum* infection led to a downregulation of the majority of the genes of the patulin biosynthetic cluster, which were instead upregulated when the fungus was inoculated alone, as already discussed (Fig. 4; Supplementary Data 13). In a recent study, we found that the BCA *P. terrestris* influences patulin production by *P. expansum* likely by secreting an unknown compound that increases the extracellular pH of the media[65], which is a key factor controlling patulin production by *P. expansum*[71]. In particular, when inoculated at high cellular concentration, the BCA *P. terrestris* led to a reduction of both the overall patulin accumulation in apples and of the specific mycotoxigenic activity (amount of patulin production per amount of *P. expansum* DNA) of *P. expansum*, which fully supports our RNAseq results showing downregulation of the *P. expansum* genes of the patulin biosynthetic cluster in the presence of the BCA.

As it concerns the transcriptomic response of the host *M. domestica*, overall there are more DEGs during the interaction with *P. expansum* than with *P. terrestris*, indicating a more robust transcriptional activation by the apple in response to the fungal pathogen. This likely reflects the different trophic relationship established by these microorganisms with the host, with *P. expansum* being a necrotrophic pathogen that infect the host to kill the tissues with invading hyphae and with the production of an arsenal of pathogenicity factors (e.g., gluconic acid, effectors, cell wall degrading enzymes, etc.), whereas *P. terrestris* establishes a non-pathogenic relationship. Several comparisons between the different datasets available were carried out, allowing to delineate both the common and the specific response of the host *M. domestica* to the BCA and to the fungal pathogen. Analysis of the apple response to the BCA aimed at assessing whether *P. terrestris* is able to induce host resistance, which is a known mechanism of several yeast BCAs[72]. First, we noticed that there is only a minor response of the host that is specific to the BCA, with the majority of the apple DEGs detected being in common with those found in response to the pathogen (Fig. 5b, c). Second, during dual interaction, the BCA led to downregulation of several biosynthetic host pathways, including those playing a role in plant defense such as those of flavonoids and ethylene (Supplementary Data 18). Third, key genes involved in plant immunity are not highly represented among the host DEGs detected in response to the BCA, but, as described later in the text, they are exclusively upregulated by the host when the pathogen *P. expansum* is present (Fig. 5b; Fig. 6; Supplementary Data 15, 16). The *M. domestica* response to both the BCA and the pathogen involves genes with diverse roles, from response to abiotic stresses, played for example by the ethylene-responsive transcription factor *ERF098* involved in salt tolerance[73], to plant resistance and activation of plant immunity, with the upregulation of the RPP13-like protein 1 that confers resistance to downy mildew[74], of the Rpm1 protein that elicits a resistance response against *Pseudomonas syringae*[75], of the WRKY DNA-binding protein 75 that positively regulates jasmonate (JA)-mediated plant defense against necrotrophic fungal pathogens[76], and of the ethylene-responsive transcription factor *ERF019* that acts as a negative regulator of PTI against *B. cinerea* and *Pseudomonas syringae*[77]. Besides these known examples, the dataset includes disease resistance proteins, cysteine-rich RLK (RECEPTOR-like protein kinase), and several heat shock proteins β glucosidases, all predicted to be involved in plant immunity but whose role has yet to be demonstrated. Moreover, studies on the BCA *P. laurentii* (previously *Cryptococcus laurentii*), which is closely related to *P. terrestris* used in this study, found that its application stimulated β-1,3-glucanase expression in jujube fruit[78], and the activity of β-1,3-glucanase, phenylalanine ammonia lyase, peroxidase, and polyphenol oxidase in pears[79]. In our dataset we only found overexpression of two predicted polyphenol oxidases (encoded by MD05G1320500 and MD05G1320800), hence only partially corroborating the aforementioned studies (Supplementary Data 14). Moreover, in the

common dataset we found: i) upregulation of the biosynthetic pathway of phenylethanol (Supplementary Data 16), which has antimicrobial proprieties against several bacteria[80] and fungi[81], is considered the active molecule responsible for the biocontrol activity of *Kloeckera apiculata* against *Penicillium* molds[82], and it has been also successfully used to extend the shelf-life of strawberries by inhibiting the growth of postharvest fungi[81]; and ii) downregulation of the pathway of the brassinosteroids (Supplementary Data 19), which are steroid phytohormones that control plant growth and development[83], are involved in protection against abiotic and biotic stresses[84], and in jujube fruits inhibited the development of blue mold rot caused by *P. expansum* through the induction of defense-related enzymes, such as phenylalanine ammonia-lyase, polyphenoloxidase, catalase and superoxide dismutase[85]. While it is reasonable to hypothesize that phenylethanol is produced by the apple tissues in response to a microbial stimulus regardless of its pathogenic potential and could be part of the induction of host defense by the BCA, downregulation of the brassinosteroids was not expected, and is in contrast with previous data on *P. expansum*-susceptible *M. domestica* cultivar Royal Gala[86]. All together these observations suggest that the induction of host defense is a mechanism of action of the yeast BCA *P. terrestris* to counteract *P. expansum* attacks on stored apples, although it likely plays a minor role compared to competition for nutrients, it involves several proteins whose roles is not defined yet in plant immunity, and it does not rely on the classical known proteins involved in PTI and ETI. Conversely, a number of characteristic genes of the plant immunity are upregulated by the host only if *P. expansum* is present, regardless of the presence of the BCA (Fig. 5b; Fig. 6).

Indeed, the data obtained showed that *P. expansum* treatment triggers the activation of the PTI, effector-triggered immunity ETI, and phytohormone signaling (Figs. 5b and 6). We found upregulation of the receptor *BAK1*, its downstream MAPK signaling cascade that involves *MEKK1-MKK1/2-MPK4-MKS1*, and the target transcription factor *WRKY33* (Fig. 6). The same cascade was found to be active also in apple and in *Arabidopsis* infected with *B. cinerea*[31,87,88], confirming that this signaling cascade is important for host response to necrotrophic pathogens. We also have evidences of activation of the chitin signaling immunity involving *CERK1*, a specific chitin receptor, and *BIK1* that phosphorylates and activates *CNGC2/4* and *RbohD* to regulate chitin-induced Ca2+ influx and ROS burst, respectively[89]. Moreover, our analyses also found activation of the jasmonic acid (JA) pathway, also known for its critical role in immunity against necrotrophic pathogens[29]. Upregulation of the JA biosynthetic pathway was found also in the study of Barad et al.[90], which compared to our study, reported also enrichment of other metabolic pathways such as that of mevalonate, flavonoids, glutathione, chorismate, and others; these pathways were also found in our study, but did not pass the threshold of FDR < 0.05, which might reflect the different *P. expansum* strain used, or the different experimental conditions. Moreover, JA coordinates the defense response against necrotrophic pathogens with ethylene, with several genes found overexpressed in our dataset (Fig. 6). We also found overexpression of genes involved in SA biosynthesis and signaling, although in general SA pathway is known to confer resistance to (hemi-) biotrophic pathogens, but can be also manipulated by necrotroph pathogens to promote disease[91]. Besides these genes involved in defense response, we also found downregulation of photosynthesis, and biosynthesis of starch and chlorophyll, which is a strategy to free up resources for the defense response and protect the photosynthetic machinery from oxidative damage[92], and is in agreement with previous findings of Naets et al. on apples infected with *B. cinerea*[88].

In conclusion, ours is one of the first studies using a trascriptomic approach to gain insight into the molecular mechanisms involved in the interaction between biocontrol agent yeast, fungal pathogen, and host, in vivo, during their active tritrophic interaction. Several other studies were carried out, but they all focused on dual interactions host-pathogen, host-BCA, and BCA-pathogen[13,63,64,93,94]. We hope that the results of our study will be pivotal to exploit the potential of the yeast BCAs, while at the same time decreasing the virulence of the fungal pathogens and increasing the resistance of the host. Techniques for analysis of gene function are present both

in fungi and plants, and can be used to confirm the function of the genes identified in the present study.

## Materials and methods
### Microbial strains and conditions for RNA sequencing

*Papiliotrema terrestris* strain PT22AV is the BCA used in the present study, and *P. expansum* strain 7015 is the fungal pathogen. Both microrganisms have been isolated by the Plant Pathology Laboratory (Department of Agricultural, Environmental and Food Sciences, University of Molise, Campobasso)[4,5]. Biocontrol experiments were carried out as reported[3]: *P. terrestris* cells were prepared by growing the yeast overnight in 20 mL of YPD (Yeast extract 10 g L$^{-1}$; Peptone 20 g L$^{-1}$; Dextrose 20 g L$^{-1}$) at 28 °C and 130 rpm, then the cells were collected, washed twice with sterilized distilled water, and adjusted at the desired cellular concentration with a hemocytometer. For *P. expansum*, conidia were collected from a two-week-old PDA (Potatoes Dextrose Agar) plate, suspended in sterilized distilled water, and adjusted at the required concentration. Apples cultivar Golden Delicious were purchased from local market. The apples were washed with commercial sodium hypochlorite at 1% for 3 min, and rinsed two times with distilled water. Once dried, four wounds of 0.2 mm of depth and 2 mm of diameter were made on each apple using a sterilized corkborer. For each wound, 30 μL of cellular suspensions of the BCA prepared as above were inoculated; after about 30 min, 15 μL of conidial suspensions of *P. expansum* prepared as above were inoculated. Apples were incubated at 24 °C, and subjected to microscopic observation every 6 h.

The biocontrol assays for RNA extraction and sequencing were carried out using a cellular concentration of the BCA at $1 \times 10^7$ CFU mL$^{-1}$, and of $2 \times 10^4$ conidia mL$^{-1}$ for the pathogen. Both the BCA and the pathogen were applied alone and in combination within artificial apple wounds, and samples for RNA extraction were collected after 36 h of incubation.

### RNA sequencing

Total mRNA from *P. terrestris* and *P. expansum* samples was isolated using a TRIzol protocol according to the manufacturer's instructions[95], while mRNA from apple tissues (both treated with the BCA and the pathogenic fungus and untreated) was extracted using an innuSPEED Plant RNA kit (analytikjena – Biometra). RNA was treated with TURBO DNase according to manufacturer's instructions, and resuspended in 50 μl of nuclease free water. RNA quality was assessed spectrophotometrically. RNA sequencing was performed at Genomix4life (Salerno, Italy) using 150 bp paired-end sequencing with Illumina Hiseq 2500. RNA-seq libraries were prepared using the TruSeq mRNA PCR-free library kits following manufacturer's instructions.

Reads obtained were subjected to quality control and bioinformatic analysis. The reads were trimmed using the software BBDuk v35.85 (http://jgi.doe.gov/data-and-tools/bb-tools/) and then were mapped against the genome assemblies with the program STAR[96]. The *P. terrestris* reference genome used was recently published by our group (GCA_019857685.1)[97]. The *M. domestica* reference genome was the GDDH13v1.1 (Phytozome genome ID: 491; NCBI taxonomy ID: 3750)[98]. The *P. expansum* reference genome was that of the strain CMP-1 available on the Ensembl database (GCA_000769755)[17]. For the *P. expansum* CMP-1 strain, the Pannzer2 pipeline was used to generate a functional gene ontology annotation because it was not available[99]. Gene expression values (counts) for each sample were obtained with FeatureCounts[100]. To identify differentially expressed genes (DEGs), the software HTSFilter[16] was first used to remove genes expressed at low levels. The quality of the biological replicates was analyzed by principal component analysis (PCA), and the reads obtained from samples relative to the dual or tritrophic interaction were mapped against the respective control condition, for a total of 7 comparisons listed below.

In particular, DEGs of *P. terrestris* were identified by comparing:
1. Apple + *P. terrestris* versus *P. terrestris* in YPD
2. Apple + *P. terrestris* + *P. expansum* versus *P. terrestris* in YPD
   DEGs of *P. expansum* were identified by comparing:
3. Apple + *P. expansum* versus *P. expansum* in PDB
4. Apple + *P. terrestris* + *P. expansum* versus *P. expansum* in PDB
   DEGs of the host *M. domestica* were identified by comparing:
5. Apple + *P. terretris* versus uninoculated apple
6. Apple + *P. expansum* versus uninoculated apple
7. Apple + *P. terrestris* + *P. expansum* versus uninoculated apple

For DEGs selection, the software EdgeR[101] was used. As a first step, all genes with a false discovery rate (FDR) < 0.05 and with a log2FC (fold change) ±1 were selected; subsequently, results within each reference organism were compared with each other using the package VennDiagram[102] resulting in additional datasets that include: for *P. terrestris* PT22AV, genes in common in the comparisons 1 and 2 reported above; for *P. expansum* 7015, genes in common in the comparisons 3 and 4 reported above; for the host *M. domestica*, genes in common in the comparison 5, 6 and 7 reported above. As a last step, a further filtering step was carried out to select as DEGs those having a log$_2$FC ± 2. The function of the *P. terrestris* and *P. expansum* DEGs was inferred by BLASTp against *S. cerevisiae* (*Saccharomyces* Genome Database, SGD, https://www.yeastgenome.org), and by gene ontology (GO) using the Pannzer2 pipeline[99]; GO accessions obtained were used for GO enrichment analysis using the topGO package (https://bioconductor.org/packages/release/bioc/html/topGO.html), and the p-value was converted into enrichment score applying the negative logarithm function. As further analysis, the GO accessions were also annotated using the KEGG orthology database[103] with pathway enrichment analysis carried out using Clusterprofiler[104].

For *M. domestica*, the function of the DEGs was inferred by BLASTp against *Arabidopsis thaliana* genome (https://www.arabidopsis.org). For *M. domestica* the GO accessions of the DEGs were obtained from the Rosaceae Database Genome (GDR) database[105]. Subsequently, GO and pathway analysis were obtained by using the AGI (Arabidopsis Genome Initiative) locus identifier[106] that provides information on the gene function and that was uploaded in the web tool Plant MetGenMAP[107], together with the fold change values. Rstudio (version 3.6.2 and version 4) was used for dataset manipulation, with the package ggplot2 used to generate the primary figures, which were assembled in the final ones using Adobe Illustrator. The flow charts were created with BioRender.

For *P. terrestris* PT22AV and *P. expansum* 7015, DEGs encoding for transporters were identified from comparison with other fungi and from the genome annotation. Their predicted function was inferred by comparison with *S. cerevisiae*, NCBI, or fungidb. Both for *P. terrestris* PT22AV and *P. expansum* 7015, DEGs encoding for transporters were displayed as a heatmap with hierarchical clustering based on their expression values, and then manually annotated for their function. For the *M. domestica* dataset, genes potentially involved in plant immunity were identified from the genome annotation based on the scientific literature[29] and on keyword search. For ETI were also selected genes annotated as resistance protein for the presence of a nucletotide binding site receptor (NBS-LRR). The heatmaps were generated with Rstudio (version 4) using the package ComplexHeatmaps.

### Growth of *Papiliotrema terrestris* PT22AV in different nitrogen sources

*Papiliotrema terrestris* PT22AV was grown overnight in shaking culture at 30 °C in YPD medium. Subsequently, the cellular pellet was washed twice with sterile water, and the cellular concentration adjusted at $1 \times 10^7$ CFU/mL using a hemocytometer. Nitrogen sources used were the 20 amino acids, ornithine, GABA, NAD, uridine, urea, allantoin, ammonium sulphate, sodium nitrite, and sodium nitrate; they were all resuspended in Minimal Medium (MM: Potassium dihydrogen phosphate 1 g L$^{-1}$; Potassium chloride 0.5 g L$^{-1}$; Magnesium sulphate 0.5 g L$^{-1}$; Dextrose 15 g L$^{-1}$) at 10 mM, with the exception of sodium nitrite that was used also at 1 mM. Moreover, an artificial apple medium (200 gr apple extract L$^{-1}$) was also used, as well as rich medium YPD as control condition. Twenty microliters of *P. terrestris* PT22AV were inoculated into 130 μl of MM, arrayed in 96 well plates and incubated at 30 °C in shaking culture for 48 h. Every 30 min

the growth was calculated using the $OD_{600}$ using a TECAN Pro Infinite instrument.

### Time course RT-qPCR analysis of *P. terrestris* genes encoding nutrient transporters identified through RNAseq analysis

Quantitative real-time PCR (RT-qPCR) analysis was performed from *P. terrestris* grown in vivo and in vitro to evaluate the expression of 10 predicted nutrient transporters (*OPT1, PUT4, THI73, MAL31, FUI1, HOL1, PTR2, GAP1, MEP2,* and *OPT2*) identified through RNAseq analysis. *Papiliotrema terrestris* target genes were subjected to BLASTn against the *Malus domestica* genome, and primers were designed on the gene regions with low or null homology, and on the exon-intron splicing junctions where possible (Supplementary Table 2).

In vivo samples were prepared by inoculating 30 μL of $1 \times 10^7$ CFU mL$^{-1}$ of the BCA *P. terrestris* in apple wounds, and after 12, 24, 36, 48, and 72 h of incubation 60 wounds (about 5 g of treated apple tissues corresponding to 250 mg of lyophilized apple tissues) were withdrawn for mRNA extraction. Untreated apple samples were included as control. In vitro samples were prepared by inoculating 30 μL of $1 \times 10^7$ CFU mL$^{-1}$ in 25 mL of apple medium and liquid YPD, and after 12, 24, 36, 48, and 72 h of incubation samples were centrifuged, the supernatant discarded, and the *P. terrestris* cellular pellet used for mRNA extraction. RNA was extracted and purified as described for RNAseq analysis.

1 μg of purified RNA was converted into cDNA via the OneScript® Plus cDNA Synthesis Kit (abm). Approximately 500 pg of cDNA were utilized to measure the relative expression level of target genes through RT-qPCR using the BlasTaq™ qPCR MasterMix (abm) in an Applied Biosystems 7900HF Real-Time PCR System. For in vivo samples, because from the mapping results it is known that RNA of *P. terrestris* in the mix "BCA + apple" is about 10% of the total (Supplementary Table 1), the amount of total cDNA used in qPCR reactions was adjusted to have the same amount of *P. terrestris* cDNA as the comparative condition in liquid culture (YPD). A control without template cDNA was included for each target. Three technical replicates were performed for each sample. Gene expression levels were normalized using the endogenous reference gene *ACT1* for in vivo samples, and *TDH1* for in vitro samples, and determined using the comparative ΔΔCt method using YPD as the comparative condition. cDNA from untreated apple wounds was used to assess *P. terrestris* primers specificity using a ΔCt approach with the treated apple samples.

### Aminoacids extraction and quantification from *P. terrestris*-treated and untreated apple wounds

Apple wounds were inoculated as described for RT-qPCR, and samples for analysis were withdrawn at 24, 36, 48, 72, and 144 h of incubation. Samples were treated by acid hydrolysis as follows: 250 mg of lyophilized apple was hydrolyzed with 20 mL of 6 N HCl at 110 °C for 24 h. The hydrolyzed samples were filtered, washed with ultrapure water, evaporated and then redissolved in 0.1 N HCl. All samples were tenfold diluted with ultrapure water and analyzed by an ICS6000 chromatographic system (Thermo Fisher Scientific S.p.A, Milano, Italy). Separation was performed with an Aminopac PA10 analytical column (250 × 2 mm, 8.5 μm particle size) (Thermo Fisher Scientific S.p.A, Milano, Italy). The chromatographic separation of the amino acids was performed using as eluents ultra-pure water, 250 mM NaOH and 1 M NaOAC with a flow rate of 0.25 mL/min using conditions shown in Supplementary Table 3.

### Statistics and reproducibility

Graphs and tables were generated using Microsoft Word, Microsoft excel, GraphPad Prism (version 9.5), and Rstudio (version 3.6.2 and version 4) script for specific software as indicated in the relative section within Material and Methods.

For RNAseq analysis, three apples each containing four wounds were used for each treatment and considered as one biological replicate. *Papiliotrema terrestris* and *P. expansum* grown for 36 h in vitro in liquid YPD and PDB, respectively, and uninoculated wounded apples wounds kept for 36 h, were used as controls. Each sample included three independent biological replicates, with the exception of the uninoculated apple and the *P. terrestris* in liquid culture that had two biological replicates, for a total of 16 samples. DEGs were identified using the software EdgeR using a FDR < 0.05 and with a $\log_2 FC \pm 2$.

For the growth kinetics of *P. terrestris* PT22AV, data shown represent the mean of five independent experiments (biological replicates) and are displayed as a heatmap with hierarchical clustering; the heatmap was generated with Rstudio (version 4) using the package ComplexHeatmaps.

For the qPCR and aminoacid quantification, 60 apple wounds (about 5 g of treated apple tissues corresponding to 250 mg of lyophilized apple tissues) were used for each time point and represented one biological replicate. In qPCR, one biological replicate and three technical replicates were carried out, while for amino acids extraction and quantification, we had two biological replicates, each consisting of one technical replicate[108].

### Reporting summary

Further information on research design is available in the Nature Portfolio Reporting Summary linked to this article.

### Data availability

Illumina raw reads for all processed samples were deposited in the NCBI SRA database under accession number PRJNA1066275 and are available at the following URL: https://www.ncbi.nlm.nih.gov/bioproject/PRJNA1066275/. In particular, data for *P. terrestris* grown in vitro are available under BioSample SAMN39484414 with accessions SRX23274217 and SRX23274218; data for *P. expansum* grown in vitro are available under BioSample SAMN39484415 with accessions SRX23274219-SRX23274221; data for *M. domestica* are available under BioSample SAMN39526954 with accessions SRX23343820 and SRX23343821 for uninoculated apple, accessions SRX23343822, SRX23343829 and SRX23343830 for apple inoculated with both the BCA and the fungal pathogen, accessions SRX23343823-SRX23343825 for apple inoculated with the fungal pathogen *P. expansum*, and SRX23343826-SRX23343828 for apple inoculated with the BCA *P. terrestris*.

The source data for Fig. 1b, c, d are in Supplementary Data 1, Supplementary Data 2, and Supplementary Data 3, respectively.

The source data for Fig. 3b, c, d are in Supplementary Data 10, Supplementary Data 11, and Supplementary Data 12, respectively.

The source data for Fig. 5b are in Supplementary Data 14 and 15, and the source data for Fig. 5c are in Supplementary Data 18 and 19.

The source data for Supplementary Fig. 5a and 5b are in Supplementary Data 1 and 2, respectively, and the source data for Supplementary Fig. 5c and 5d are in Supplementary Data 6.

The source data for Supplementary Fig. 7a and 7b are in Supplementary Data 8 and 9, respectively, and the source data for Supplementary Fig. 7c and 7d are in Supplementary Data 12.

The source data for Fig. 2, Fig. 4 and Fig. 6, and for Supplementary Fig. 6a, 6c and 6d, Supplementary Fig. 9, Supplementary Fig. 11, and Supplementary Fig. 12 are in Supplementary Data 20.

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

## Acknowledgements

This work was supported by the PON AIM program Azione I.2 Attrazione e Mobilità dei Ricercatori (to G.I. and R.C.), and by the PRIN 2022 PNRR P2022TPZW3 to G.I.

## Author contributions

G.I.: Experiments conceptualization, experimental design, data analysis, data interpretation, supervision, funding acquisition, manuscript writing. G.B.: Data analysis. D.P.: RNAseq set-up and RNA extraction. M.Q. and I.G.: aminoacids extraction and quantification. F.D.C.: critical revision of the manuscript. R.C.: Experiments conceptualization, supervision, funding acquisition and manuscript revision.

## Competing interests

The authors GI, DP, FdC and RC are co-authors of an international application for the biocontrol agent used in the present study, *P. terrestris* strain PT22AV (international application number PCT/IB2022/053387). All other authors declare no competing interests. The strain PT22AV of *P. terrestris* is the propriety of the Company Agroventure that has the rights to distribute it.
