## [Peer Review File · Communications Biology]

Reviewers' comments:

Reviewer #1 (Remarks to the Author):

The MS submitted by Ianiri et al., described a extensive transcriptomic analysis between the biocontrol PT22AV, the fungal pathogen *P. expansum* and the host *M. domestica*. In my point of view, the work is well performed and described. The information will be of the interest for the scientist working in this field.

I have only few comments to be consider.

1. Line 297 and 300. Information is not presented and labeled as "Data not shown". I consider that is important to present the data as supplementeray information.
2. Line 322 to 331. This information should be on Materials and Methods.
3. Across the text tables are written as "table X". Should be in capital letter "Table X".
4. Figures are not mentioned in the text immediatly after the results are described. For example: Figure 2D is part of the first section (line 359) and is mentioned only on line 453. Please check for all the figures.

Reviewer #2 (Remarks to the Author):

The manuscript by Ianiri and co-authors reports a thorough and detailed RNAseq analysis of the tritrophic interaction between the biocontrol yeast *Papiliotrema terrestris*, the fungal plant pathogen *Penicillium expansum*, and the apple host (*Malus domestica*). The authors performed competition experiments with these three organisms and compared transcriptome data with results obtained from the control experiments (organisms alone or only interaction between two organisms). A number of genes specifically up- or downregulated in the yeast, the pathogen or the host are identified and discussed in detail.

Overall, this is a comprehensive transcriptomic analysis of a yeast biocontrol interaction that hints at the mechanisms and crosstalk of biocontrol interactions. The study reports a rich source of data for future studies of mechanisms and will certainly be welcome by the field.

A few specific comments and suggestions for changes are the following (numbers refer to line numbers):

Abstract: Maybe divide into sections (on lines 38, 44, and 50) and introduce the abbreviations PTI and ETI. It might be worthwhile adding a few numbers to the abstract (DEGs in different conditions and organisms).

70: Yeasts are fungi – maybe specify as “filamentous fungi”.

93: “its” instead of “is”.

104: Maybe “biocontrol research” instead of “the biocontrol”.

292ff: 300000 yeast cells or BCA cells but not yeast cells of the BCA.

296: Past tense (did). In several instances in the results section the authors should rather use past than present tense (e.g., when referring to the groups of genes that they found).

312: Maybe pathogen instead of fungus?

336ff: *in vitro* in italics.

364ff: Maybe specifically state “135 DEGs” instead of genes. In general (also in other sections, the numbers seem a bit confusing and difficult to follow. Maybe it would be easier to present the actual number of DEGs and not only those specific for a condition. Or it must be clearly stated that the number only refers to the DEGs that are unique for a specific condition.

374f: Suggestion: In table 1, the most are reported.

378 and reference to figures/figures in general: Figure 3b is cited before 2d and 3a. The Figures should be arranged in the order they appear in the text.

376: An upregulated phosphate transporter is interesting! Is phosphate limiting under these conditions? (could be tested by adding phosphate to the interaction and observing the differences).

504: polyamine instead of polyamine.

532: Should it not be xyloglucanases?

562: hydrolyze instead of hydrolize.

775/777: It is not necessary to spell out the genus names again.

922: Suggestion: ... excluded that some of them have a direct

Results in general: The description of the results is very detailed and thorough, which is good. On the other hand, it is also quite lengthy and somewhat difficult to follow and read. Maybe, if the journal permits, it would be possible to introduce additional subheadings. The separate sections for the BCA, pathogen and host are helpful, but the key findings could be highlighted by specific sub-sections with a corresponding sub-title.

Figure 1: Maybe it would be possible to arrange it with 3 horizontal pictures and 2 rows. Headings (18h, 24h, 36h and *P. expansum* and *P. terrestris*) could be included on top and the left of the photographs.

Figure 2: In panel b, the overlap of the two circles does not represent the numbers. The overlap should be much larger and only small areas should be on the left and right. Should the overlap not be green? The labels in Panel c are very small and not readable. Maybe panel b could be moved to the top right and c below, and be larger. Panel d only appears later in the text.

Figure 3: Good size and readable. In the text, B is mentioned before A. Maybe omit A and B and label the two columns in the figure.

Figure 4: Same as for Figure 2.

Figure 6: The labels in panels b and c are very difficult to read.

Figure 7: Also very difficult to read and too small. Maybe it should cover a whole page. Also please add the panels labels in the figure.

Reviewer #3 (Remarks to the Author):

The manuscript “Transcriptomic approach to unveil the interaction between a biocontrol yeast and postharvest fungal pathogen on the host fruit” shows transcriptomic analysis of the BCA *P. terrestris*

strain PT22AV, the pathogenic *P. expansum* and the host *M. domestica*. The authors undertook detailed analysis of RNAseq for the interaction of host with pathogen and/or BCA as well as analysed interaction of all 3. This study contributes to a better understanding of the molecular basis underlying the interaction between biocontrol and host. This is crucial in order to achieve implementing biocontrol options in agriculture. The authors identified a range of DEGs in the different scenarios, in particular plant defense responsive, amino acid metabolism and carbohydrate transporters were differentially expressed. Although many of the findings were, as mentioned in the manuscript itself, already identified by previous studies, a novel aspect was shown by using this particular BCA strain and analysing tritrophic interactions. However, I think there are some aspects that could improve this work:

- 1) Is there evidence for biocontrol activity of *P. terrestris* in *M. domestica*? Did the authors perform phenotypic analysis of apples incubated with and without *P. terrestris* and with and without the addition of *P. expansum*? Were there any changes in disease development observed? This would be useful and a quick experiment to establish if any biocontrol activity exists.
- 2) It is a shame that only one timepoint was chosen for the RNAseq analysis. I do realise that there might have been financial reasons for that but an earlier timepoint (such as 12 or 18 hours) would give additional information of transcriptional changes that are happening early after colonisation. There might have been plant specific responses early on during the interaction that are not present or less expressed in the later timepoint. Same might be true for the yeast and fungus, some transcriptional regulations might have occurred very early after exposure to the other microbe.
- 3) This study claims major translational shifts are happening during the interactions. I realise that this might be out of the scope of this study but polysome profiling or RiboSeq analysis that are focussing on changes on the ribosome level would be interesting next steps to further reveal the molecular mechanisms involved.
- 4) There is evidence presented that *P. terrestris* influences amino acid metabolism, but the authors did not perform follow-up experiments that would further strengthen these findings. Free amino acids can be measured relatively easily by GCMS and an experiment designed to look at apple tissue and the amino acids available in the presence and absence of *P. terrestris* would give further insights on whether *P. terrestris* uptakes amino acids or possibly the host plant might synthesize higher amounts of amino acids to support growth of *P. terrestris*.
- 5) I am missing follow up experiments confirming high or low expression of identified DEGs. The authors mention that most of the identified DEGs were already previously found and therefore would be validated but they did identify some genes that were not previously published. This is the novel aspect of their study, so they claim, but no simple follow up experiments such as qPCR was performed to confirm expression of some of the nutrient transporters for example. I think confirming some of the high and low expressed genes in the different interaction groups is crucial as based on the settings in the data analysis, results may differ.

6) Some parts in the result sections, in particular information about data analysis could be moved to methods section.

Reviewers' comments:

Reviewer #1 (Remarks to the Author):

The MS submitted by Ianiri et al., described a extensive transcriptomic analysis between the biocontrol PT22AV, the fungal pathogen *P. expansum* and the host *M. domestica*. In my point of view, the work is well performed and described. The information will be of the interest for the scientist working in this field.

Response: we thank the reviewer 1 for evaluating our manuscript.

I have only few comments to be consider.

1. Line 297 and 300. Information is not presented and labeled as "Data not shown". I consider that is important to present the data as supplementeray information.

Response: we have deleted the "Data not shown" as requested since we realized that it was not necessary.

2. Line 322 to 331. This information should be on Materials and Methods.

Response: We disagree with the reviewer. In Materials and Methods (lines 257-270) we described the methods used to analyze the data, while in the Results (lines 447-456) we do a brief recap and explain the advantage of the methods used for the analysis of mixed samples.

3. Across the text tables are written as "table X". Should be in capital letter "Table X".

Response: the text was modified as requested

4. Figures are not mentioned in the text immediately after the results are described. For example: Figure 2D is part of the first section (line 359) and is mentioned only on line 453. Please check for all the figures.

Response: we disagree with the reviewer. Figures contain several panels each containing a lot of information that are also complementary to the tables and the supplementary tables. There could be several ways to organize the results: one could describe every figure and table as separate blocks of information, but they could be also described together and with the correct indications in the text to orientate the reader. We believe that this last option is more appropriate and explains better the results that we obtained. In the specific case raised by the reviewer, figure 2D was moved in the text to appear before figure 3 (lines 529-533), as asked by reviewer 2. We have checked our manuscript as to all main figures are recalled in the text in the order they appear.

Reviewer #2 (Remarks to the Author):

The mansuscript by Ianiri and co-authors reports a thorough and detailed RNAseq analysis of the tritrophic interaction between the biocontrol yeast *Papiliotrema terrestris*, the fungal plant pathogen *Penicillium expansum*, and the apple host (*Malus domestica*). The authors performed competition experiments with these three organisms and compared transcriptome data with results obtained from the control experiments (organisms alone or only interaction between two organisms). A number of genes specifically up- or downregulated in the yeast, the pathogen or the host are identified and discussed in detail.

Overall, this is a comprehensive transcriptomic analysis of a yeast biocontrol interaction that hints at the mechanisms and crosstalk of biocontrol interactions. The study reports a rich source of data for future studies of mechanisms and will certainly be welcome by the field.

A few specific comments and suggestions for changes are the following (numbers refer to line numbers):

Abstract: Maybe divide into sections (on lines 38, 44, and 50) and introduce the abbreviations PTI and ETI. It might be worthwhile adding a few numbers to the abstract (DEGs in different conditions and organisms).

Response: we agree with the reviewers but unfortunately for this journal the abstract cannot have sections, and it has a very limited number of allowed words (200) so it is difficult to include additional information; however, we followed almost all the suggestions of reviewer 2, with the exception of the sections which are not allowed by the journal, and the indication of the apple DEGs that would take a lot of space because characterized by several comparisons. Consider that when preparing the first submission we had to re-write several sentences in the abstract to shorten it to follow the journal indications (in the resubmitted Article the deleted abstract can be seen).

70: Yeasts are fungi – maybe specify as “filamentous fungi”.

Response: changed as requested.

93: “its” instead of “is”.

Response: changed as requested.

104: Maybe “biocontrol research” instead of “the biocontrol”.

Response: changed as requested.

292ff: 300000 yeast cells or BCA cells but not yeast cells of the BCA.

Response: changed as requested.

296: Past tense (did). In several instances in the results section the authors should rather use past than present tense (e.g., when referring to the groups of genes that they found).

Response: changed as requested. We modified also other parts of the text as suggested by the reviewer 2.

312: Maybe pathogen instead of fungus?

Response: replaced fungus with fungal pathogen

336ff: *in vitro* in italics.

Response: changed as requested.

364ff: Maybe specifically state “135 DEGs” instead of genes. In general (also in other sections, the numbers seem a bit confusing and difficult to follow. Maybe it would be easier to present the actual number of DEGs and not only those specific for a condition. Or it must be clearly stated that the number only refers to the DEGs that are unique for a specific condition.

Response: we thank the reviewer for this comment, and as suggested we replaced genes with DEGs as suggested in all cases. We believe that the text clearly indicates the specific condition, and it recalls the figure in which these numbers are clearly represented in the Venn Diagram.

374f: Suggestion: In table 1, the most are reported.

Response: changed as requested in all cases.

378 and reference to figures/figures in general: Figure 3b is cited before 2d and 3a. The Figures should be arranged in the order they appear in the text.

Response: we thank the reviewer for this comment. As indicated before, figures contain several panels each containing a lot of information that are also complementary with the tables and the supplementary tables, and it can be tricky to describe them in the order that Figures are recalled in the text. Even though we reported the KEGG analysis after the gene ontology for each condition in the entire manuscript, for this part we have modified the text as to describe figure 2D before figure 3 (lines 529-533).

376: An upregulated phosphate transporter is interesting! Is phosphate limiting under these conditions? (could be tested by adding phosphate to the interaction and observing the differences).

Response: we thank the reviewer for this comment. This is definitely true, our analysis suggest that phosphate is important, but also many other nutrients are important. The present study will serve as a guide for future experiments and analyses; we do appreciate the suggestion of the reviewer and will evaluate the role of phosphate and other nutrients in the upcoming experiments.

504: polyamine instead of poliamine.

Response: changed as requested.

532: Should it not be xyloglucanases?

Response: yes, we changed accordingly.

562: hydrolyze instead of hydrolize.

Response: changed as requested.

775/777: It is not necessary to spell out the genus names again.

Response: changed as requested.

922: Suggestion: ... excluded that some of them have a direct

Response: changed as requested.

Results in general: The description of the results is very detailed and thorough, which is good. On the other hand, it is also quite lengthy and somewhat difficult to follow and read. Maybe, if the journal permits, it would be possible to introduce additional subheadings. The separate sections for the BCA, pathogen and host are helpful, but the key findings could be highlighted by specific sub-sections with a corresponding sub-title.

Response: we thank the reviewer for this comment. We agree that subheadings are helpful for the reader, and have indeed added them in several parts of the result section.

Figure 1: Maybe it would be possible to arrange it with 3 horizontal pictures and 2 rows. Headings (18h, 24h, 36h and P. expansum and P. terrestris) could be included on top and the left of the photographs.

Response: we thank the reviewer for this comment. Because the reviewer is only suggesting a different arrangement of the panels without modifying its content, we added a legend to the figure for a better understanding, but we prefer keeping it as in the first submission.

Figure 2: In panel b, the overlap of the two circles does not represent the numbers. The overlap should be much larger and only small areas should be on the left and right. Should the overlap not be green? The labels in Panel c are very small and not readable. Maybe panel b could be moved to the top right and c below, and be larger. Panel d only appears later in the text.

Response: we thank the reviewer for this comment. Venn diagrams can be drawn in relation to the numbers they report (ie being larger where a higher number is reported), or can be drawn in an equal way regardless on the numbers they report. We decided to use this last option because it works better for some representation as in some cases one number is very small compared to the other (eg 43 in figure 2, or Venn diagrams of figure 7) and that does not look nice. The overlap of the Venn in figure 2B represents common genes as the mid column of figure 2C and it is in a dark green indeed. We printed a high-quality image of the figure and we checked that the text is actually readable. Moving the panels as the reviewer suggests would result in a figure that is larger than 1 page, and we don't like that. The text corresponding to figure 2D has been moved earlier in the text (lines 529-533) to reflect the order they are recalled, as suggested earlier by reviewer 2.

Figure 3: Good size and readable. In the text, B is mentioned before A. Maybe omit A and B and label the two columns in the figure.

Response: we thank the reviewer for this comment. Figure 3 is not divided in A and B as these letters only indicates the different treatments, and we corrected the text and recall the figure as Figure 3.

Figure 4: Same as for Figure 2.

Response: we thank the reviewer for this comment. What was figure 4 in the first submission now is figure 5. As mentioned in our response to the comment to figure 2, it is better to represent the Venn diagram in a constant way in all the figures. We are aware that the labels of the GO in figure 5C are small, but they are readable in a high-quality printed image; it is impossible to make the text bigger and include all the information.

Figure 6: The labels in panels b and c are very difficult to read.

Response: we thank the reviewer for this comment. What was figure 6 in the first submission now is figure 7. As mentioned in our earlier responses the labels of the GO are readable in a high-quality printed image; it is impossible to make the text bigger and include so many information.

Figure 7: Also very difficult to read and too small. Maybe it should cover a whole page. Also please add the panels labels in the figure.

Response: we thank the reviewer for this comment. What was figure 7 in the first submission now is figure 8. We have modified the figure as suggested, so it covers an entire page now. The panel labels are many characters and take a lot of space, further reducing the size of the images. For this reason, they are recalled in the legend.

Reviewer #3 (Remarks to the Author):

The manuscript "Transcriptomic approach to unveil the interaction between a biocontrol yeast and postharvest fungal pathogen on the host fruit" shows transcriptomic analysis of the BCA *P. terrestris* strain PT22AV, the pathogenic *P. expansum* and the host *M. domestica*. The authors undertook detailed analysis of RNAseq for the interaction of host with pathogen and/or BCA as well as analysed interaction of all 3. This study contributes to a better understanding of the molecular basis underlying the interaction between biocontrol and host. This is crucial in order to achieve implementing biocontrol options in agriculture. The authors identified a range of DEGs in the different scenarios, in particular plant defense responsive, amino acid metabolism and carbohydrate transporters were differentially expressed. Although many of the findings were, as mentioned in the manuscript itself, already identified by previous studies, a novel aspect was shown by using this particular BCA strain and analysing tritrophic interactions. However, I think there are some aspects that could improve this work:

1) Is there evidence for biocontrol activity of *P. terrestris* in *M. domestica*? Did the authors perform phenotypic analysis of apples incubated with and without *P. terrestris* and with and without the addition of *P. expansum*? Were there any changes in disease development observed? This would be useful and a quick experiment to establish if any biocontrol activity exists.

Response: we thank the reviewer for this comment. Evidences of the biocontrol activity of *P. terrestris* against several pathogens are present since 1997, when the first work from our lab on *P. terrestris* was published (See Castoria et al., 1997; reference 4 in the manuscript). Since then, our lab published several reports about the biocontrol activity of *P. terrestris*, and these can be seen in the text and in the references (see references 3, 4, 5, 6, 7, which are the most relevant that support this manuscript); following a specific request by the editor, we included more published work on *P. terrestris* and discussed in more details the key findings of them (lines 131-156). We have very well characterized the biocontrol activity of *P. terrestris* against *P. expansum* on apple, and use this as a study model to understand the mechanisms of action of the BCA. It would have been a very naive approach performing such a massive RNAseq analysis (in terms of both data analysis and financial resources used) without knowing whether *P. terrestris* antagonizes

P. expansum on apple! We think that this request of the reviewer is redundant with our previous work (the most recent published in AEM PMID: 33452020 - Molecular Tools for the Yeast *Papiliotrema terrestris* LS28 and Identification of Yap1 as a Transcription Factor Involved in Biocontrol Activity) as it does not add novel information and makes the manuscript even longer, considering also the addition of further experiments requested by reviewer 3 (see below).

2) It is a shame that only one timepoint was chosen for the RNAseq analysis. I do realise that there might have been financial reasons for that but an earlier timepoint (such as 12 or 18 hours) would give additional information of transcriptional changes that are happening early after colonisation. There might have been plant specific responses early on during the interaction that are not present or less expressed in the later timepoint. Same might be true for the yeast and fungus, some transcriptional regulations might have occurred very early after exposure to the other microbe.

Response: We thank the reviewer for this comment. We agree with the reviewer, an earlier time point as well as a later time point will for sure provide additional information. But at some point a researcher has also to be rationale and tailor experiments to available fundings (they are always limiting in normal labs). Nevertheless, we do believe that our study provides much novel and original information that paves the way to further experimental activity and to fostering the practical implementation of biocontrol in agriculture, as reviewer 1 and reviewer 2 pointed out, and also reviewer 3 herself/himself noticed as to the analysis of transcriptome of the tritrophic interaction.

3) This study claims major translational shifts are happening during the interactions. I realise that this might be out of the scope of this study but polysome profiling or RiboSeq analysis that are focussing on changes on the ribosome level would be interesting next steps to further reveal the molecular mechanisms involved.

Response: We thank the reviewer for this comment. We believe that this is an interesting approach that we will consider in the future, but it is indeed out of scope in the present manuscript.

4) There is evidence presented that *P. terrestris* influences amino acid metabolism, but the authors did not perform follow-up experiments that would further strengthen these findings. Free amino acids can be measured relatively easily by GCMS and an experiment designed to look at apple tissue and the amino acids available in the presence and absence of *P. terrestris* would give further insights on whether *P. terrestris* uptakes amino acids or possibly the host plant might synthesize higher amounts of amino acids to support growth of *P. terrestris*.

Response: We thank the reviewer for this comment. As requested, we have now performed analyses of the amino acids in apple tissues treated with *P. terrestris* at different time points, and compared that to amino acids analysis carried out on untreated apple tissues. The experimental set up followed that of the RNAseq analysis, with the only difference being the treatment with the pathogen *P. expansum* that was not performed. In particular, a number of apple wounds were treated with the BCA *P. terrestris*, and 60 wounds (about 5 grams of treated apple tissues that correspond to 250 mg of lyophilized apple tissues) were withdrawn at 24, 36, 48, 72 and 144 hours after inoculation for amino acids extraction and analysis. Unfortunately, the results obtained showed that there were no statistical differences between the untreated and treated apple wounds for most samples, with the exception of threonine and serine+ proline that were higher at 144 hpi; there is a clear reduction of aspartic acid in apple sample treated with the BCA at 144 hpi, but it did not reach statistical significance (Figure 1 of the present file).

While doing this experiment we also realized that the experimental set up we used for RNAseq analysis does not work well to determine amino acids utilization by *P. terrestris* in vivo, as explained below in the representation in figure 2 of the present file: when placed onto apple wounds, the yeast BCA does not have the ability to invade the fruit tissue like a pathogenic fungus, but it rather remains on the wound surface (panel A). As a consequence, the withdrawn wounds to be analyzed included also untreated apple fruit tissue around and below the inoculated wound (panels B and C, respectively), and hence most likely not exposed to the BCA metabolism. A similar approach for amino acid analysis has been used for the pathogen *P. expansum* by Gong et al. 2022 (PMID: 36359980 – Dynamic Change of Carbon and Nitrogen Sources in Colonized Apples by *Penicillium expansum*), but in that case the apple tissue to be analyzed it is easily identified as it is the rotten tissue (panel D).

Figure 2: experimental set up for withdrawn apple samples for amino acids and real time qPCR analyses

Last, in the results obtained we even noticed a different amino acids content in different untreated apples, which makes it very difficult to appreciate whether the BCA might utilize low amount of certain amino acids, further confirming that the experimental set up used for RNAseq analysis (and qPCR, see next point) is not very good to evaluate in vivo the utilization of aminoacids by the BCA *P. terrestris*.

Because the experiment was done to address request by the reviewer 3, we believe that it is worth to include these data, **and inserted them in lines 363 – 374 (Materials and Methods), lines 970 – 983 (Discussion), and represented as Figure S8 and Table S2 in the new version of the manuscript**; please note that this analysis has been performed by two additional PhD students that have now been included as authors of the manuscript.

Last, we also tried to quantify amino acids in a liquid apple medium as done for qPCR (see next point), but unfortunately the extraction did not work well. Although the use of liquid medium could reduce variability as compared to the apple samples, it is clear that we need to develop a specific protocol for amino acids extraction.

5) I am missing follow up experiments confirming high or low expression of identified DEGs. The authors mention that most of the identified DEGs were already previously found and therefore would be validated but they did identify some genes that were not previously published. This is the novel aspect of their study, so they claim, but no simple follow up experiments such as qPCR was performed to confirm expression of some of the nutrient transporters for example. I think confirming some of the high and low expressed genes in the different interaction groups is crucial as based on the settings in the data analysis, results may differ.

Response: We thank the reviewer for this comment. As requested, also this other experiment has been performed. Samples were prepared as done for amino acids analysis, and withdrawn at 12, 24, 36, 48, and 72 hours after inoculation.

This was not a simple qPCR as stated by reviewer 3 because the RNA to be converted in cDNA is extracted from samples consisting of 2 organisms (BCA and apple), and therefore several extra precautions have to be taken into account to achieve a reliable analysis. Of course the same was done for RNAseq analysis, but it is clear that the principles behind a RNAseq analysis and a qPCR are quite different, as the first relies on the use of algorithms that try to map with the highest specificity long Illumina reads (150 bp paired-end in our case) on assembled genome sequences, whereas the second relies on short primers (18 – 22 nucleotides) that have to anneal specifically to a region of the *P. terrestris* genome in a mixed sample in which the amount of *P. terrestris* RNA extracted is much lower than the total RNA; as determined in the RNAseq analysis by the percentage of reads from 'BCA + apple' samples that mapped to the corresponding genomes (see supplemental table 3 of the new version of the manuscript), the RNA of *P. terrestris* in the mix 'BCA + apple' is about 10% of the total, the remaining is apple RNA. Therefore, a high amount of total cDNA has to

be used in qPCR reaction in order to have the equal amount of *P. terrestris* cDNA as the control condition in liquid culture (YPD). Moreover, this high amount of apple cDNA might interfere with the specificity of the primers designed for *P. terrestris* genes, and to reduce this problem, all *P. terrestris* target genes chosen were searched by BLASTn in the *Malus domestica* genome, aligned, and primers were designed on the regions with low or null homology and on the exon-intron splicing junctions where possible. The specificity of primers for conserved housekeeping genes was expected to be problematic and, for this reason, two *P. terrestris* genes were tested [actin (*ACT1*) and glyceraldehyde-3-phosphate dehydrogenase GAPDH (*TDH1*)].

To assess the interference of the apple cDNA with the primers for *P. terrestris* genes, qPCR reactions were performed on the cDNA of the untreated apple sample at the concentration to be used in reaction considering the 10% of the *P. terrestris* cDNA mentioned above. Based on the CT values obtained (see table 2 below), all the primers showed a certain degree of amplification against the apple cDNA, with the lowest specificity that was obtained for the housekeeping gene *TDH1*, and for the genes *PTR2*, *MEP2*, *OPT2* and *GAP1*.

Genes	Mean CT values obtained in qPCR with the untreated apple using primers specific for P. terrestris
ACT1	35.515854
TDH1	21.969355
OPT1	32.4282225
PUT4	35.760419
THI73	30.926035
MAL31	33.444633
FUI1	37.536564
HOL1	35.7940215
PTR2	22.3929
GAP1	28.0806615
MEP2	25.6626415
OPT2	27.259673

Table 2: Mean CT values obtained in qPCR with the untreated apple using primers specific for *P. terrestris*.

Subsequently, all the primers were tested also against all the treated samples 'BCA + apple' and the log2FC expression calculated using a ΔCT approach with the 'apple sample alone' as reference condition (figure 3 of the current file).

A positive value means that the expression of the gene in the treated sample ('BCA + apple') is higher than the expression of the same gene in the apple alone (untreated host). The housekeeping gene *TDH1* was the only one that showed negative values, which means that its expression was higher in the untreated host and therefore it cannot be used as reference gene using the $\Delta\Delta\text{CT}$ formula. **This graph has been combined with the table 2 and included in the manuscript as supplemental figure S5.**

A classical qPCR was performed using the $\Delta\Delta CT$ formula with the reference condition being *P. terrestris* grown in YPD, and with the reference gene being *ACT1*. The results are shown in the following graph (figure 4).

Time course real time qPCR carried out for 10 upregulated predicted nutrient transporters of the BCA showed that the majority of them are upregulated from 12 hpi to 72 hpi; RNAseq analysis revealed higher expression for most selected DEGs, with the exception of *OPT1*, *GAP1*, *MEP2* and *OPT2*. **These data have been inserted in the manuscript as new figure 4A.**

Although we found overexpression of the majority of the genes selected, hence overall validating the RNAseq data, to exclude the influence of the apple cDNA we carried out a time course experiment of *P. terrestris* grown in apple medium (200 gr apple extract L⁻¹) and withdrawn samples for RNA extraction at 12, 24, 36, 48, and 72 hours after inoculation. In this case the gene *TDH1* showed more stable expression during the entire experiment compared to *ACT1*, and hence used as the reference housekeeping gene.

Also using an apple medium, we confirmed overexpression of all the DEGs selected, with the exception of *PTR2* at early time points (Figure 5 of the present file). Moreover, there are some differences with the expression data obtained using

the inoculated wounded apples, and these differences might be due to the technical interference of the host cDNA with *P. terrestris* primers. **These data have been inserted in the manuscript as new figure 4 B.** In summary, data relative to the qPCR have been inserted in the manuscript in lines 333-361 (Materials and Methods), lines 627-649 (Results), lines 959-964 (Discussion), and represented in figure 4 (A and B), supplemental figure S5, and supplemental table S1.

6) Some parts in the result sections, in particular information about data analysis could be moved to methods section.
Response: Reviewer 3 has been not very specific in this point raised, and we hope that the modifications that we made are acceptable.

REVIEWERS' COMMENTS:

Reviewer #1 (Remarks to the Author):

I consider that MS should be published in the current version.

Reviewer #2 (Remarks to the Author):

As already stated in my first review of this manuscript, I believe that this is a comprehensive piece of work that presents a wealth of data that will be a welcome resource and interesting for the community working on biocontrol yeasts.

My specific comments have all been addressed and incorporated. In general, I have the impression that the authors invested a lot of effort in this revision; in particular, they now present new qRT-PCR data and amino acid analyses. In the latter analyses, the authors (unfortunately as they state) did not detect differences between treated and untreated samples for most amino acids. Although more or stronger differences would certainly have been nice, at least in this reviewer's opinion, these results do not exclude that amino acid transport is involved in the biocontrol interaction against plant pathogenic fungi (which, as far as I understand, were not included in these experiments). I do not suggest additional experiments, just a maybe less discouraging conclusion.

Reviewer #3 (Remarks to the Author):

The authors have dedicated considerable time and effort to conduct supplementary experiments, which effectively reinforce and enhance the findings derived from the RNA sequencing analysis. I commend the authors for their diligence in implementing these additional experiments, and I believe that these efforts have significantly elevated the overall quality of the manuscript. The authors have successfully addressed all of my previously raised concerns, and I find no further comments to be made.